# Selective suppression of melanoma lacking IFN-γ pathway by JAK inhibition depends on T cells and host TNF signaling

Hongxing Shen[1,8], Fengyuan Huang[2,8], Xiangmin Zhang[3], Oluwagbemiga A. Ojo[1], Yuebin Li[1], Hoa Quang Trummell[1], Joshua C. Anderson[1], John Fiveash[1,4], Markus Bredel[1,4], Eddy S. Yang [1,4], Christopher D. Willey [1,4], Zechen Chong [2,4] ✉, James A. Bonner [1,4,9] ✉ & Lewis Zhichang Shi [1,4,5,6,7,9] ✉

Therapeutic resistance to immune checkpoint blockers (ICBs) in melanoma patients is a pressing issue, of which tumor loss of IFN-γ signaling genes is a major underlying mechanism. However, strategies of overcoming this resistance mechanism have been largely elusive. Moreover, given the indispensable role of tumor-infiltrating T cells (TILs) in ICBs, little is known about how tumor-intrinsic loss of IFN-γ signaling (IFNγR1KO) impacts TILs. Here, we report that IFNγR1KO melanomas have reduced infiltration and function of TILs. IFNγR1KO melanomas harbor a network of constitutively active protein tyrosine kinases centered on activated JAK1/2. Mechanistically, JAK1/2 activation is mediated by augmented mTOR. Importantly, JAK1/2 inhibition with Ruxolitinib selectively suppresses the growth of IFNγR1KO but not scrambled control melanomas, depending on T cells and host TNF. Together, our results reveal an important role of tumor-intrinsic IFN-γ signaling in shaping TILs and manifest a targeted therapy to bypass ICB resistance of melanomas defective of IFN-γ signaling.

ICBs such as anti-CTLA-4 and anti-PD-1/L1 induce unprecedented clinical benefits in patients with various types of advanced cancer and are revolutionizing the field of cancer treatment[1–3]. Over the past decade or so, more than 70 approvals have been granted to ICBs by the FDA[2–7], some of which are for first-line use, establishing ICBs as a major pillar of cancer care. Notwithstanding these transformative clinical successes, the overall efficacy of ICBs is limited to a small subset of cancer patients due to frequently encountered therapeutic resistance[8]. Using a cohort of advanced melanoma, we found that ~75% of melanoma patients did not respond to anti-CTLA-4 therapy and their tumors harbored losses of IFN-γ signaling genes[9]. Similar findings were reported for anti-PD-1 therapy[10] and subsequently corroborated by a series of seminal studies in melanoma and colon cancer[11–14]. Together, these studies reveal that tumor loss of IFN-γ signaling is a major mechanism of resistance to ICBs[9–14]. However, therapeutic approaches to overcome this ICB resistance have remained largely unknown.

ICBs, by blocking immune checkpoints (namely, CTLA-4, PD-1, and PD-L1) hijacked by tumor cells to evade immunosurveillance, enhance the effector function (e.g., IFN-γ production)[15,16] and decrease the abundance of immunosuppressive FoxP3+ regulatory T cells (Treg) in TILs[17], leading to tumor rejection. In support of this, we found that the interactive loop of IFN-γ and IL-7 signaling in T cells dictates the

[1]Department of Radiation Oncology, Heersink School of Medicine, University of Alabama at Birmingham (UAB-SOM), Birmingham, AL 35233, USA. [2]Department of Genetics and Informatics Institute, UAB-SOM Birmingham, AL, USA. [3]Department of Pharmaceutical Sciences, Wayne State University, Detroit, MI 48201, USA. [4]O'Neal Comprehensive Cancer Center, UAB-SOM Birmingham, AL, USA. [5]Department of Microbiology, UAB-SOM Birmingham, AL, USA. [6]Department of Pharmacology and Toxicology, UAB-SOM Birmingham, AL, USA. [7]Programs in Immunology, UAB-SOM Birmingham, AL, USA. [8]These authors contributed equally: Hongxing Shen, Fengyuan Huang. [9]These authors jointly supervised this work: James A. Bonner, Lewis Zhichang Shi. ✉e-mail: zchong@uabmc.edu; jabonner@uabmc.edu; Lewisshi@uabmc.edu

therapeutic efficacy of anti-CTLA-4 and anti-PD-1[18]. Although both T cell- and tumor-intrinsic IFN-γ signaling are required for ICB response, surprisingly, our original characterizations of TILs isolated from melanomas with knockdown of the essential IFN-γ receptor 1 (IFNγR1[KD]) did not reveal overt changes of the CD8[+]/T[reg] ratio[9], a commonly used index of TILs' effector function. While this suggests that tumor IFN-γ signaling may not impart TILs, a caveat is that IFNγR1[KD] melanoma still has residual IFN-γ signaling and is not an ideal model to assess how the loss of IFN-γ signaling in tumor cells modulates TILs.

In this study, to circumvent the partial attenuation of IFN-γ signaling in IFNγR1[KD] melanoma and to unequivocally evaluate how tumor IFN-γ signaling affects TILs, we generate the B16 melanoma model with *Ifngr1* knocked out by CRISPR-Cas9 (hereafter, IFNγR1[KO]). In contrast to IFNγR1[KD] melanomas, IFNγR1[KO] melanomas show a reduced abundance of CD8[+] T cells at the baseline and lack increased infiltration and functional rejuvenation of TILs upon anti-CTLA-4 therapy. Bioinformatic analyses of human melanomas with impaired IFN-γ signaling also reveal reduced expression of T cell signature genes. Interestingly, our multi-omics studies inform a network of constitutively active PTKs centered on activated JAK1/2, downstream of the heightened mTOR signaling pathway in IFNγR1[KO] cells. In direct correlation, human melanomas with reduced IFN-γ signaling or ICB resistance exhibit upregulation of target genes in the mTOR and JAK1/2 pathways, indicative of their activation. Targeting activated JAK1/2 with Ruxo selectively

suppresses IFNγR1[KO] but not scrambled control melanomas, coupled with enhanced effector functions (e.g., TNF production) and reduced T[reg] frequency in TILs. Subsequently, deletion of T cells and host TNF signaling completely abolish therapeutic effects of Ruxo, highlighting an indispensable role of T cells and host TNF signaling in this process. Collectively, we demonstrate that tumor-intrinsic IFN-γ signaling actively regulates infiltration and function of TILs; our results support Ruxo as a potential "targeted" therapy for ICB-resistant IFNγR1[KO] melanoma. Since Ruxo is clinically approved, this study may lead to a rapid repurposing of Ruxo to treat melanomas lacking IFN-γ signaling.

## Results

### Creation of a "clean" melanoma model lacking IFN-γ signaling

Our previous work using the syngeneic IFNγR1[KD] melanoma model identified a theretofore unreported role of tumor-intrinsic IFN-γ signaling in anti-CTLA-4 response[9]. However, IFNγR1[KD] melanoma still retained some degree of IFN-γ signaling, evidenced by significant upregulation of inducible PD-L1 by IFN-γ (Supplementary Fig. 1a, the right panel), preventing us from explicitly assessing how tumor loss of IFN-γ signaling modulates TILs and ICB response. To circumvent this, we created the IFNγR1[KO] B16-BL6 melanoma model using CRISPR-Cas9 technology (Fig. 1a). Unlike IFNγR1[KD] cells, IFNγR1[KO] cells were completely resistant to IFN-γ stimulation, indicated by the lack of IFN-γ-induced p-JAK2 (Fig. 1b), no transcriptional upregulation of *Irf1*

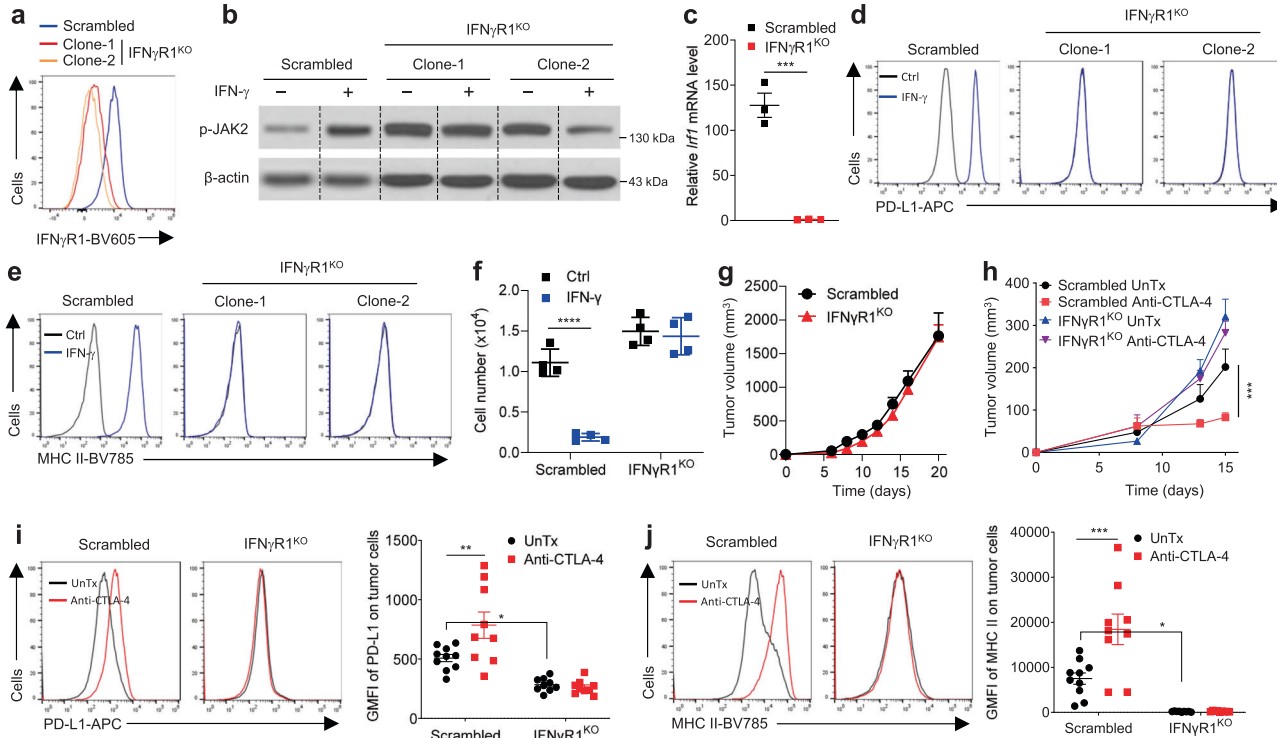

**Fig. 1 | Generation and characterization of IFNγR1[KO] melanoma model lacking functional IFN-γ signaling.** B16-BL6 cells were transduced with specific single guide RNAs (sgRNAs) against exon #1 of mouse *Ifngr1* or scrambled sgRNAs. **a** IFNγR1 expression in scrambled control and IFNγR1[KO] clones by flow cytometry (FACS strategy 1). **b** p-JAK2 in scrambled control and IFNγR1[KO] clones untreated (UnTx) or treated with IFN-γ (100 U/mL for 15 min) by Western blot. β-actin was the loading control. Experiments were repeated twice with similar results. **c** mRNA expression of *Irf1* in scrambled control (*n* = 3) and IFNγR1[KO] cells (*n* = 3) treated with 1000 U/mL of IFN-γ for 90 min. ***p = 0.0007 by two-sided Student's *t*-test. **d–f** Scrambled control and IFNγR1[KO] cells were untreated (UnTx) or treated with 100 U/mL IFN-γ for 24 h to detect surface expression of PD-L1 (**d**) and MHC II (**e**) by flow cytometry (FACS strategy 1), or for 48 h to count live cells (**f**) (*n* = 4 per group). ****p = 0.00005 (Ctrl vs IFN-γ groups for the same cell type), by two-sided Student's

*t*-test. **g** Tumor growth of scrambled control (*n* = 5) and IFNγR1[KO] (*n* = 5) melanomas in Rag-1[−/−] mice. **h** Tumor growth of scrambled control and IFNγR1[KO] melanomas in B6 mice treated with anti-CTLA-4 or isotype control (UnTx). *N* = 5 for Scrambled UnTx/Anti-CTLA-4; *n* = 4 for IFNγR1[KO] UnTx/Anti-CTLA-4. ***p = 0.0007, by two-way ANOVA with Dunnett's multiple comparisons test (with adjustment). **i, j** Surface expression of PD-L1 (**i**) and MHC II (**j**) on isolated tumor cells (CD45[−]) by flow cytometry (FACS strategy 3). *N* = 10 for Scrambled UnTx, *n* = 9 for each of the other three groups. In **i**, *p = 0.0415 and **p = 0.0082; in **j**, *p = 0.0283 and ***p = 0.0006, by two-way ANOVA with Tukey's multiple comparisons test (with adjustment). Representative data from two independent experiments are shown. The scatter plots and line graphs depict means ± SEM. Source data are provided in the Source Data file.

(a direct downstream target of IFN-γ signaling, Fig. 1c), as well as no upregulation of PD-L1 (Fig. 1d), MHC II (Fig. 1e), and MHC I (Supplementary Fig. 1b). Furthermore, IFN-γ did not induce overt cell death in IFNγR1[KO] cells, assessed by 7-AAD and Annexin V staining (Supplementary Fig. 1c), neither did it suppress cell proliferation, indicated by no dilution of CellTrace Violet (CTV, a cell proliferation dye) (Supplementary Fig. 1d). Consequently, total numbers of viable IFNγR1[KO] cells were not reduced, contrasting a drastic decrease of scrambled control cells in response to IFN-γ (Fig. 1f).

To examine whether IFNγR1[KO] affected tumor formation in vivo, we inoculated Rag-1[−/−] mice lacking mature T and B cells with scrambled control and IFNγR1[KO] cells. Consistent with a previous report showing comparable growth of melanomas lacking other important genes in the IFN-γ signaling[13], we did not observe overt growth defect of IFNγR1[KO] melanoma (Fig. 1g). We also did not find altered growth kinetics of IFNγR1[KO] tumor in immunocompetent B6 mice, in the absence of ICBs (Fig. 1h). In keeping with reported ICB resistance in tumors with impaired IFN-γ signaling[9–14], IFNγR1[KO] melanomas did not respond to anti-CTLA-4 treatment and continued to grow, whereas scrambled control melanomas were suppressed by anti-CTLA-4 (Fig. 1h). In line with our in vitro data, direct analyses of IFNγR1[KO] tumor cells (CD45[−]) did not show upregulation of PD-L1 (Fig. 1i) and MHC II (Fig. 1j) upon anti-CTLA-4, in contrast to marked upregulation in scrambled control melanoma cells. In aggregate, IFNγR1[KO] melanomas lack functional IFN-γ signaling and are completely resistant to ICBs and IFN-γ stimulation, presenting a "clean" system to interrogate how tumor-intrinsic loss of the IFN-γ signaling imparts TILs.

## Reduced infiltration and function of TILs in IFNγR1[KO] melanoma

In line with an essential role of tumor IFN-γ signaling in tumor antigen presentation[14], we noticed a drastic reduction of MHC molecules in IFNγR1[KO] cells (Fig. 1j), suggesting an inefficient process of T cell cross-priming in IFNγR1[KO] melanomas. However, our previous analysis of IFNγR1[KD] melanomas did not unveil altered ratios of CD8[+]/T_{reg}[9], a widely accepted indication of TILs' function. Considering IFNγR1[KD] melanomas still possessed IFN-γ signaling (albeit weaker) (Supplementary Fig. 1a), we revisited this issue by analyzing TILs isolated from the "clean" IFNγR1[KO] melanomas. Appallingly, unlike IFNγR1[KD] melanomas[9], IFNγR1[KO] melanomas had markedly reduced CD8[+] T cells at the baseline and no increased T cell infiltration upon anti-CTLA-4 therapy, as compared to scrambled control melanomas (Fig. 2a). In addition, anti-CTLA-4 failed to deplete intratumoral T_{reg} (Fig. 2b), did not increase the CD8[+]/T_{reg} ratio (Fig. 2b), and did not promote the production of effector cytokines by CD8[+] (Figs. 2c and S2b) and CD4[+] TILs (Supplementary Fig. 2a). Increasing trends of IFN-γ production by CD8[+] (Supplementary Fig. 2c) and CD4[+] (Supplementary Fig. 2d) TILs were noticed in scrambled control but not IFNγR1[KO] melanomas upon anti-CTLA-4. Similarly, anti-CTLA-4 increased the expression of T cell activation marker PD-1 on both CD8[+] (Fig. 2d) and CD4[+] TILs (Supplementary Fig. 2e) and concurrently reduced CD73 expression, an immunosuppressive ectoenzyme that catalyzes immunostimulatory ATP to potent immunosuppressive adenosine[19], only in scrambled control melanoma. Taken together, these data indicate that melanomas with dysfunctional IFN-γ signaling have reduced infiltration and function of TILs, pointing to an important role of tumor IFN-γ signaling in shaping TILs.

We previously reported that patients with advanced melanoma harboring loss of IFN-γ signaling genes were resistant to anti-CTLA-4 therapy[9]. However, how IFN-γ signaling in human melanomas regulates TILs has not been reported. Inspired by our preclinical findings, we posited that human melanomas with attenuated IFN-γ signaling would have reduced expression of T cell signature genes, including prototypical surface markers for T cells (CD3, CD4, and CD8), effector molecules (IFNG, GZMB, perforin (PRF1), and TNF), and MHC molecules (MHC I: HLA-A, HLA-B, and HLA-C; MHC II: HLA-DRA). Since our

previously published database[9] was derived from whole exome sequencing and did not contain gene expression data, we were unable to address this using that dataset. To circumvent this, we first analyzed the TCGA database of human skin cutaneous melanomas (SKCMs) (n = 458). Specifically, we grouped SKCMs into IFNGR1[High] vs IFNGR1[Low] using the median expression of IFNGR1 in melanoma cells after deconvolution of the bulk samples with a panel of melanoma-specific genes[20]. We reasoned that IFNGR1[Low] SKCMs would have attenuated IFN-γ signaling and thus reduced expression of T cell signature genes. Indeed, we observed significantly reduced expression of CD3, CD4, CD8, HLA-DRA, GZMB, IFNG, and TNF in bulk IFNGR1[Low] SKCMs, while the others (HLA-A, HLA-B, HLA-C, and PRF1) were also reduced (although not significant) (Fig. 2e). Interestingly, in correlation with their lower T cell signature, IFNGR1[Low] SKCMs had worse survival probabilities (p = 0.0039) (Supplementary Fig. 2f), suggesting weaker anti-tumor responses in these patients. Secondly, unlike SKCMs being responsive to ICBs, uveal melanomas (UVMs) have been known to be resistant to ICBs[21]. We, therefore, assessed IFNGR1 expression in UVMs vs SKCMs after the aforementioned deconvolution and found significantly reduced IFNGR1 expression in UVMs (Fig. 2f), suggestive of weaker IFN-γ signaling in UVMs than SKCMs. Importantly, bulk UVMs also had decreased expression of most T cell signature genes (except for just one: HLA-A) (Fig. 2f). These data suggest that human melanomas with attenuated IFN-γ signaling have decreased expression of T cell signature genes, reflective of reduced T cell infiltration and function, corroborating our preclinical findings. Noteworthily, dysfunctional IFN-γ signaling (IFNγR1[KO]) is required to impart TILs in murine melanomas, as TILs in IFNγR1[KD] melanoma are largely unaltered[9]. However, in human melanomas, attenuated IFN-γ signaling as in IFNGR1[low] SKCMs and in UVMs (lower IFNGR1 expression than SKCMs) is sufficient to induce appreciable effects on TILs, implying that TILs in human melanomas are more sensitive to the dysregulation of tumor IFN-γ signaling. Despite this gradient discrepancy between murine and human melanoma, our results nevertheless highlight an important role of tumor IFN-γ signaling in shaping TILs.

## Constitutively active JAK1/2 in IFNγR1[KO] melanoma

Although tumor loss of IFN-γ signaling has been defined as a major mechanism of resistance to anti-CTLA-4 (Fig. 1h)[9] and anti-PD-1[10–14], little effort has been devoted to overcome this ICB resistance. We thus attempted to uncover therapeutic targets that can be harnessed to treat ICB-resistant melanomas lacking functional IFN-γ signaling. Considering the important role of PTKs in coordinating the IFN-γ signaling cascade, we conducted a global kinase activity analysis (kinomics). Because PTK inhibitors are readily available for pharmacological targeting, we specifically focused on activated PTKs that have positive Mean Kinase Statistics (MKS, a readout for extent and direction of change) and Mean Final Scores (MFS, indicative of specificity) greater than 0.5. Following these criteria, we found 26 activated PTKs in IFNγR1[KO] cells (Supplementary Table 1), including receptor tyrosine kinases (RTKs such as Ephrin receptor A and B (EphA/B)) as well as non-receptor tyrosine kinases (NRTKs: spleen tyrosine kinase (Syk) and ZAP70) that are known to be involved in carcinogenesis[22]. To our surprise, we also observed activated JAK1 and JAK2, essential downstream components of the IFN-γ signaling pathway[23]. More intriguingly, when these constitutively activated PTKs were integrated for annotated network modeling, a JAK1/2-centric network emerged (Fig. 3a), highlighting a central role of active JAK1/2 in the rewiring of these kinases. To directly confirm this finding, we analyzed phosphorylation of JAK1 and JAK2 (p-JAK1 and p-JAK2) by Western blot (WB) in cells cultured under normoxia (21% $O_2$) and hypoxia (1% $O_2$, mimicking hypoxic tumor microenvironment [TME]). Consistent with our kinomic data, p-JAK1 and p-JAK2 were increased in IFNγR1[KO] cells (Fig. 3b). Similarly, basal p-JAK1 and p-JAK2 were increased in IFNγR1[KD] cells (Supplementary Fig. 3a). We also assessed the three kinases (Syk,

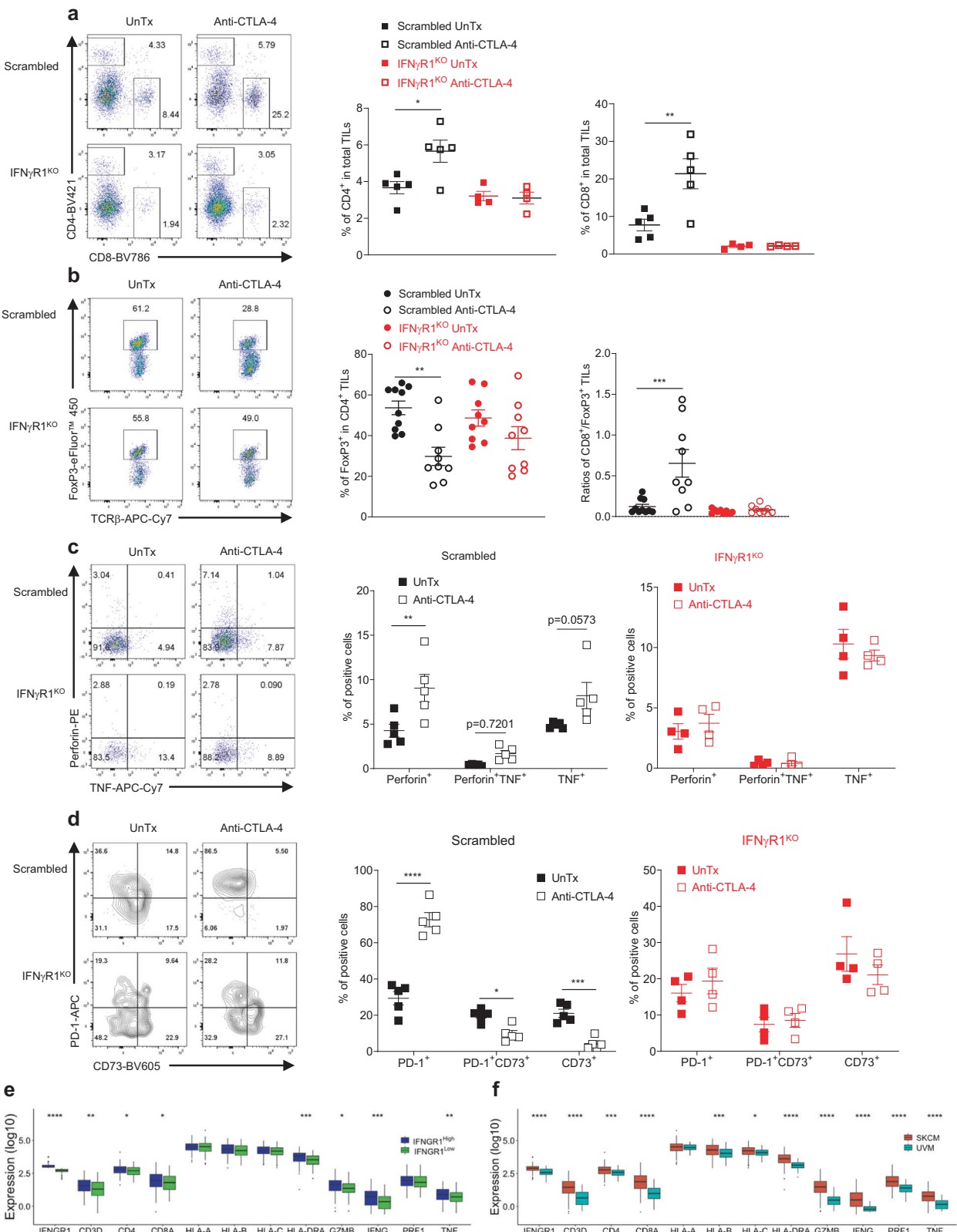

ZAP70, and EphA3) with high MFS from our kinomic study (Supplementary Table 1) by WB. Of note, basal p-Syk (Supplementary Fig 3b) and p-ZAP70 (Supplementary Fig. 3c) were very low in these cells. Although p-EphA3 was detectable (Supplementary Fig. 3c), they did not show significant increases in IFNγR1[KO] cells. Given these results and the central role of JAK1/2 in the PTK network, we dedicated our subsequent efforts on JAK1/2.

A classical downstream event of activated JAK1/2 is tyrosine phosphorylation of STATs, particularly STAT1 and STAT3[23]. We thus examined p-STAT1/3 by WB. Surprisingly, we could not detect p-STAT1, even with a substantial amount of protein loading and prolonged film exposure times, indicating a low level of basal p-STAT1 in melanoma. On the other hand, although the basal level of p-STAT3 was also low, it was detectable and increased in IFNγR1[KO] cells, suggesting

**Fig. 2 | Tumor-intrinsic IFN-γ signaling shapes tumor-infiltrating T cells.** Isolated tumor-infiltrating lymphocytes (TILs) from scrambled control and IFNγR1$^{KO}$ melanomas treated with or without (UnTx) anti-CTLA-4 were analyzed for the abundance of CD4$^+$ and CD8$^+$ T cells (**a**) (*$p = 0.0173$; **$p = 0.0044$), FoxP3$^+$ cells among CD4$^+$ TILs (**b**) (**$p = 0.0025$; ***$p = 0.0004$), TNF and perforin production by CD8$^+$ TILs after a brief stimulation with PMA and ionomycin (**c**) (**$p = 0.0043$), and surface expression of PD-1 and CD73 on unstimulated CD8$^+$ TILs (**d**) (*$p = 0.033$; ***$p = 0.0005$; ****$p = 0.00004$). The scatter plots in **a**–**d** depict representative data (means ± SEM) from two independent experiments. $N = 5$ for Scrambled UnTx/Anti-CTLA-4, $n = 4$ for IFNγR1$^{KO}$ UnTx /Anti-CTLA-4 in **a**, **c**, **d**. $N = 10$ for Scrambled UnTx, $n = 9$ for Scrambled Anti-CTLA-4, IFNγR1$^{KO}$ UnTx and IFNγR1$^{KO}$ Anti-CTLA-4 groups in **b**. One-way ANOVA with Tukey's multiple comparisons test (with adjustment) was used for statistical analyses in **a** and **b**, and two-way ANOVA with Šídák's

multiple comparisons test (with adjustment) in **c** and **d**. FACS strategy 3 was applied in **a**–**d**. **e** Skin cutaneous melanomas (SKCMs) in the TCGA database were grouped into IFNGR1$^{High}$ ($n = 101$) and IFNGR1$^{Low}$ ($n = 150$) according to IFNGR1 expression in melanoma cells (after deconvolution using a panel of melanoma-specific genes). Comparisons of T cell signature genes in the bulk (without deconvolution) IFNGR1$^{High}$ vs IFNGR1$^{Low}$ SKCMs were presented as boxplots. **f** Expression of IFNGR1 (after deconvolution) and T cell signature genes (without deconvolution) in SKCMs ($n = 251$) vs uveal melanomas (UVMs) ($n = 58$) from the TCGA database. The boxes in **e**, **f** depict the first (lower) quartile, median (center line), and the third (upper) quartile, and the vertical lines indicate the minimum and maximum values. The statistical analyses in **e**, **f** were calculated using R with Mann–Whitney $U$-test. *$p < 0.05$; **$p < 0.01$; ***$p < 0.001$; ****$p < 0.0001$. Source data as well as exact $p$ values for **e**, **f** are provided in the Source Data file.

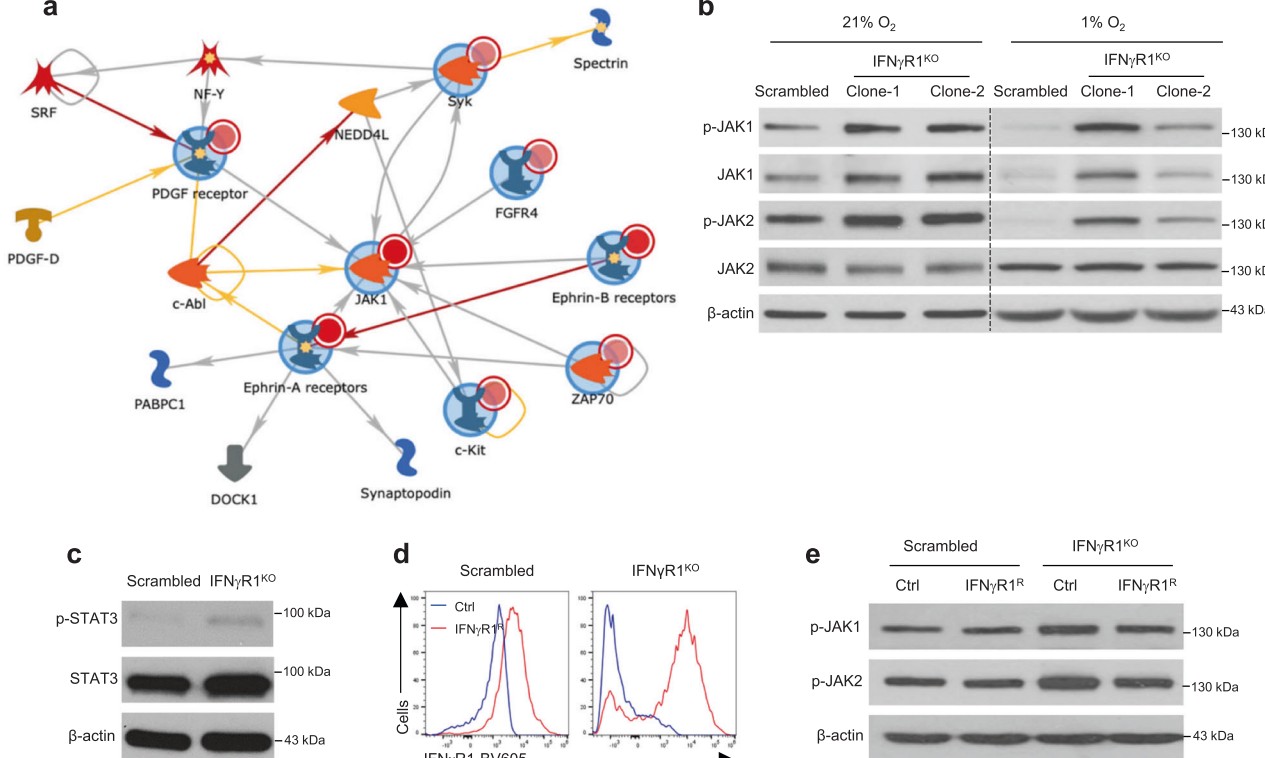

**Fig. 3 | Constitutive activation of JAK1/2 in IFNγR1$^{KO}$ melanoma cells.**
**a** Identification of a JAK1/2-centric network of activated protein tyrosine kinases in IFNγR1$^{KO}$ cells by kinomic analysis. Input nodes (kinases) with large blue circles around them and smaller red circles on the top right corner indicate increased activity in IFNγR1$^{KO}$ cells. Arrowheads denote the direction of interaction and colors of the lines indicate the type of interaction (yellow: positive; red: negative; gray: context-dependent). **b** Scrambled and IFNγR1$^{KO}$ cells were cultured under normoxic (21% O$_2$) or tumor microenvironment-mimicking hypoxic (1% O$_2$) culture

conditions, followed by Western blot (WB) analyses of p-JAK1/2 and total-JAK1/2. **c** p-STAT3 and total-STAT3 in scrambled and IFNγR1$^{KO}$ cells by WB. **d**, **e** Scrambled and IFNγR1$^{KO}$ cells were transduced with control lentiviruses (Ctrl) or lentiviruses encoding mouse IFNγR1 for re-expression (IFNγR1$^R$). Successfully transduced cells were analyzed for IFNγR1 expression by flow cytometry (FACS strategy 1) (**d**) and p-JAK1/2 by WB (**e**). β-actin was used as a loading control in WB. Experiments were repeated twice with similar results in **b**, **c**, and **e**. Source data are provided in the Source Data file.

that STAT3 is a preferential target of activated JAK1/2 in IFNγR1$^{KO}$ cells (Fig. 3c). Because we used single IFNγR1$^{KO}$ clones but not mixtures in this study to avoid interference from cells with inefficient/partial deletion of *Ifngr1* by CRISPR-Cas9, a potential concern would be that activated JAK1/2 may occur merely by chance in single clones rather than a direct outcome of deletion of IFN-γ signaling. To address this, we re-expressed *Ifngr1* in scrambled control and IFNγR1$^{KO}$ cells to comparable levels (IFNγR1$^R$) (Fig. 3d), using lentiviruses encoding mouse *Ifngr1*. Compellingly, IFNγR1$^R$ greatly reduced p-JAK1/2 in IFNγR1$^{KO}$ cells and largely rescued the overly increased p-JAK1/2 (Fig. 3e), directly linking lack of IFN-γ signaling to aberrant JAK1/2 activation in melanoma.

## JAK1/2 activation in IFNγR1$^{KO}$ melanoma is unlikely mediated by extrinsic signals

Next, we wanted to shed light on how the JAK1/2 were activated in IFNγR1$^{KO}$ cells. As we recently reviewed[23], the JAK-STAT pathway is a rapid membrane-to-nucleus signaling module regulated by a wide array of extracellular signals, including cytokines and growth hormones. In addition to IFN-γ, type I interferons such as IFN-α/β[23] and IL-6[24] are among the major extrinsic signals that engage the JAK-STAT pathway. To determine whether JAK1/2 activation in IFNγR1$^{KO}$ cells could be due to enhanced IL-6 signaling, we analyzed the expression of *Il6* and *Il6r*, both of which were significantly upregulated (Fig. 4a). To evaluate whether this enhanced IL-6 signaling mediated JAK1/2

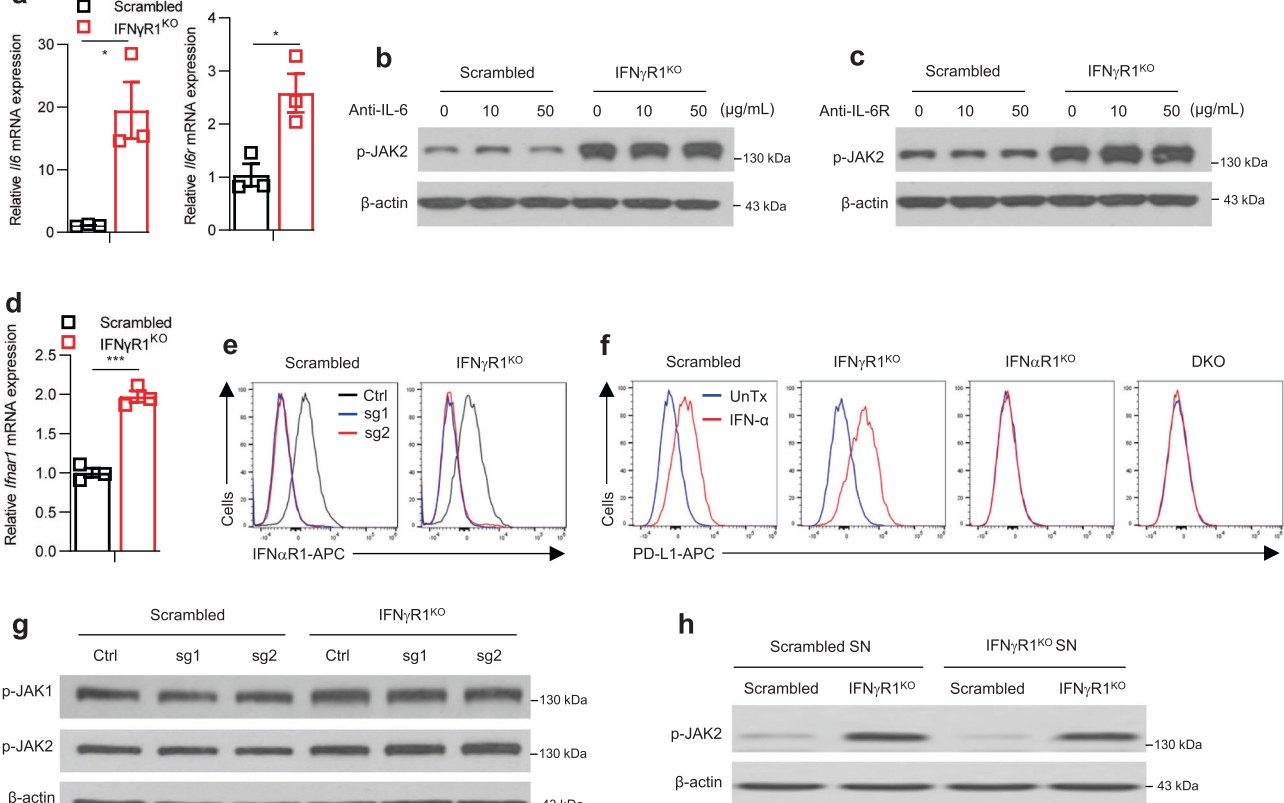

**Fig. 4 | Activation of JAK1/2 in IFNγR1ᴷᴼ cells is not mediated by extrinsic signals. a** mRNA expression of *Il6* (*\*p* = 0.0153) and *Il6r* (*\*p* = 0.0216) in scrambled control (*n* = 3) and IFNγR1ᴷᴼ (*n* = 3) cells by real-time RT-PCR. Representative data from two independent experiments are shown as means ± SEM. **b, c** p-JAK2 in scrambled control and IFNγR1ᴷᴼ cells pretreated with various doses of blocking antibodies against IL-6 (**b**) or IL-6R (**c**), analyzed by Western blot (WB). Experiments were repeated twice with similar results. **d** mRNA expression of *Ifnar1* in scrambled control (*n* = 3) and IFNγR1ᴷᴼ (*n* = 3) cells by real-time RT-PCR. Representative data from two independent experiments are shown as means ± SEM. *\*\*\*p* = 0.0005.

**e–g** Scrambled and IFNγR1ᴷᴼ cells were transduced with different sgRNAs against mouse *Ifnar1*. Successfully transduced cells were analyzed for IFNαR1 expression in untreated cells (**e**) and PD-L1 expression after stimulation with 100 ng/mL IFN-α for 48 h (**f**) by flow cytometry (FACS strategy 1) and p-JAK1/2 in untreated cells by WB (**g**). **h** p-JAK2 in scrambled and IFNγR1ᴷᴼ cells incubated with supernatants (SN) harvested from scrambled or IFNγR1ᴷᴼ cultures for 24 h, analyzed by WB. β-actin was used as a loading control in WB. All the experiments were repeated twice with similar results. A two-sided Student's *t*-test was used for statistical analyses in **a**, **d**. Source data are provided in the Source Data file.

activation, we blocked IL-6 and IL-6R with anti-IL-6 and anti-IL-6R antibodies, respectively, at concentrations that were sufficient to inhibit IL-6-induced p-STAT3 in melanoma cells (Supplementary Fig. 4a). Unfortunately, blocking IL-6 (Fig. 4b) and IL-6R (Fig. 4c) did not restore increased p-JAK1/2 in IFNγR1ᴷᴼ cells, suggesting IL-6 signaling is not involved in JAK1/2 activation.

We then asked if type I interferon signaling contributes to JAK1/2 activation. To this end, we analyzed IFNαR1, the essential receptor for IFN-α/β, and found it was significantly upregulated in IFNγR1ᴷᴼ cells (Fig. 4d). We interrogated if IFNαR1 upregulation would lead to greater IFN-α signaling. To this end, we stimulated scrambled control and IFNγR1ᴷᴼ cells with various doses of IFN-α, followed by an examination of p-STAT1 and p-STAT3, which did not show greater increases in IFNγR1ᴷᴼ cells (Supplementary Fig. 4b). Also, IFNγR1ᴷᴼ cells did not show enhanced sensitivity to IFN-α-induced killing (Supplementary Fig. 4c). While these results suggested that JAK1/2 activation in IFNγR1ᴷᴼ cells may not be due to enhanced IFN-α signaling, to explicitly rule out this, we deleted *Ifnar1* in scrambled control and IFNγR1ᴷᴼ cells using CRISPR-Cas9 with different single guide RNAs (sg1 and sg2) (Fig. 4e). We confirmed the ablation of the IFN-α signaling in these cells, evidenced by no inducible PD-L1 upregulation after IFN-α stimulation (Fig. 4f). Importantly, this ablation of IFNαR1 did not rescue JAK1/2 activation (Fig. 4g), indicating a dispensable role of IFN-α signaling in JAK1/2 activation. Lastly, to explore the potential regulation of JAK1/2 activation by other extrinsic factors secreted by IFNγR1ᴷᴼ cells

into the supernatant (SN) (cytokines, growth factors, extracellular vesicles, etc.), we treated scrambled control cells with SNs harvested from IFNγR1ᴷᴼ cultures for 24 h. This did not induce increased p-JAK2 (Fig. 4h), suggesting a nonessential role of extrinsic factors in JAK1/2 activation. Of note, increased p-JAK2 in IFNγR1ᴷᴼ cells persisted, irrespective of the SNs (IFNγR1ᴷᴼ or scrambled control) used, indicating that JAK1/2 activation is more of a cell-intrinsic event.

**Augmented mTOR pathway mediates JAK1/2 activation in IFNγR1ᴷᴼ melanoma**

In addition to extracellular signals (IFN-α, IL-6, etc.), constitutive activation of JAK1/2 can result from cell-intrinsic alterations (i.e., enhanced intracellular signaling[25]). To gain a global idea of this, we performed a whole transcriptome analysis of scrambled control and IFNγR1ᴷᴼ cells, which identified 265 downregulated genes and 332 upregulated genes (Fig. 5a). We performed a signaling pathway enrichment analysis using these differentially expressed genes (DEGs). This unsupervised analysis revealed a wide array of pathways that were significantly affected (Fig. 5b), including essential intracellular pathways in tumor aggression and therapeutic resistance (e.g., PI3K-Akt, p53, FoxO, MAPK, and mTOR pathways[26–29]), pathways important in tumor cell growth and proliferation (e.g., cell cycle[26], glutathione metabolism[30], arginine, proline metabolism[31], etc.), as well as pathways involved in the formation of various types of cancer (e.g., prostate cancer, breast cancer, colorectal cancer, melanoma, gastric cancer, etc.). This confirms a

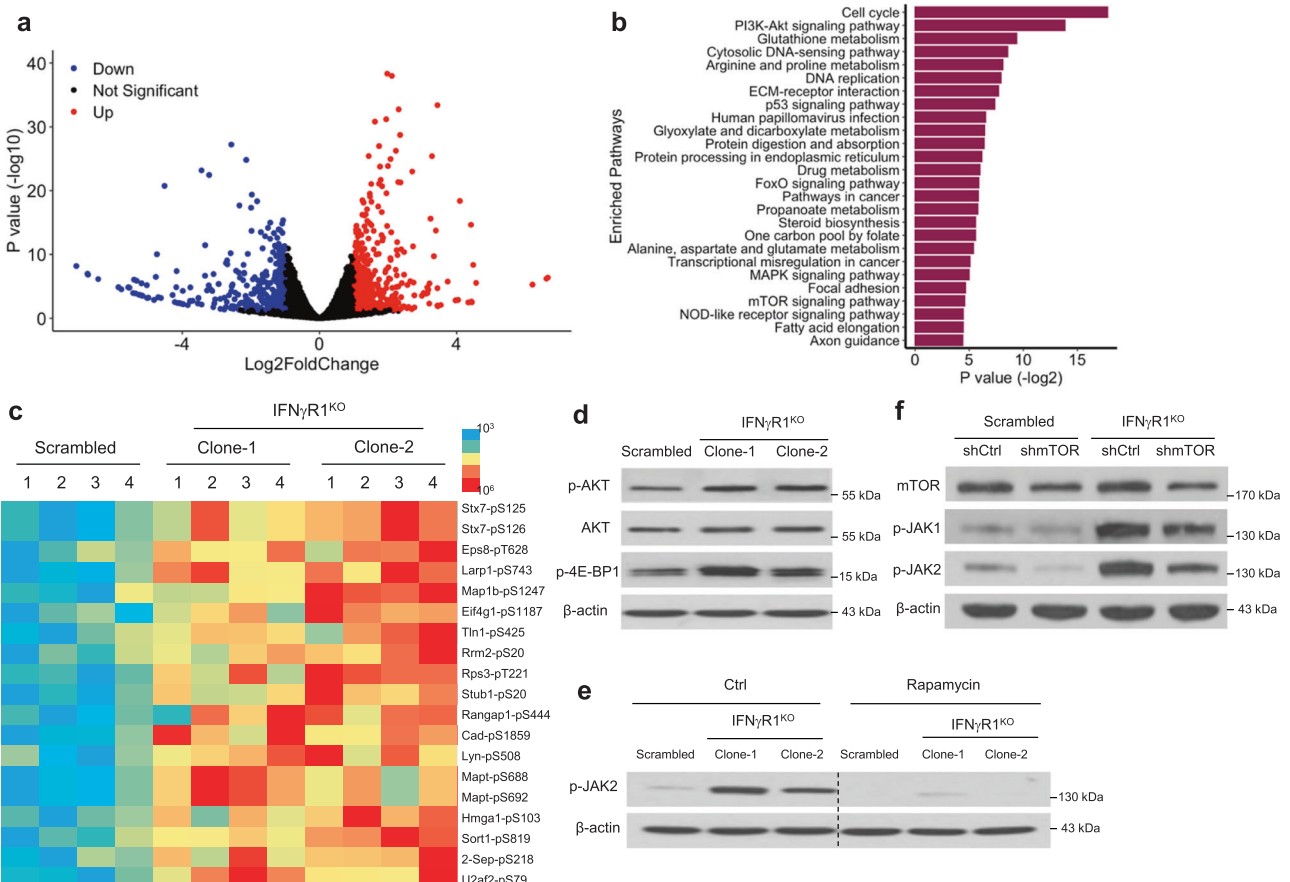

**Fig. 5 | Heightened PI3K-AKT-mTOR axis in IFNγR1^KO cells mediates JAK1/2 activation. a, b** Upregulated and downregulated genes in scrambled (*n* = 3) and IFNγR1^KO (*n* = 3) cells by RNA-Seq (**a**) and top hits of altered signaling pathways in IFNγR1^KO cells (**b**). The gene expression analyses were performed using DESeq2 (version 1.34.0). The Wald test was used to calculate the p values and log2 fold changes. Genes with an adjusted *p* value < 0.05 and absolute log2 fold change >1 were considered as differentially expressed genes (DEGs). A volcano plot was used to show all upregulated and downregulated DEGs using the ggplot2 R package. Enriched Kyoto Encyclopedia of Genes and Genomes (KEGG) pathways of the DEGs were identified by enrichr package. Significant terms of the KEGG pathways were selected with a *p* value < 0.05. **c** Cell lysates of scrambled control (*n* = 4) and

IFNγR1^KO (*n* = 4) cells were subjected to mass spectrometry-based phosphoproteomic analysis. Phosphorylation sites known to be mediated by experimentally defined kinases were shown in the heatmap. Blue and red colors indicate low and high expression levels, respectively. **d** p-AKT, total-AKT, and p-4E-BP1 in scrambled and IFNγR1^KO cells were analyzed by Western blot (WB). **e** p-JAK2 in scrambled and IFNγR1^KO B16-BL6 cells untreated (Ctrl) or pretreated with rapamycin (1 μM) for 3 h, analyzed by WB. **f** Scrambled and IFNγR1^KO cells were transduced with lentiviruses expressing nonspecific shRNAs (shCtrl) or mTOR shRNAs (shmTOR), followed by analyses of mTOR, p-JAK1/2 by WB. β-actin was the loading control in WB. Experiments were repeated twice (**f**) or thrice (**d, e**) with similar results. Source data are provided in the Source Data file.

widespread impact of IFN-γ signaling loss in tumor cells on tumor progression and therapy response, including its role in ICB resistance[9].

To directly detect the activities of these intracellular signaling pathways, we conducted phosphoproteomic studies with a special focus on serine/threonine kinases, considering their intricate interactions with PTKs[22]. Our analysis identified 7529 phosphosites, of which 217 showed significantly increased phosphorylation in IFNγR1^KO cells (deposited to massive.ucsd.edu and also included in the source data file). We paid special attention to the ones catalyzed by experimentally well-defined kinases (Fig. 5c); targeted proteins and phosphosites were listed on the right. Uniprot IDs (mouse) for these kinases were then used to map to KEGG IDs for pathway enrichment analyses, which defined 23 signaling pathways (Supplementary Table 2). Because the same phosphopeptides can be mediated by different kinases, it is rare to have definitive cognate phosphopeptides for individual kinases. We, therefore, reason that if the activation of kinases in one pathway can explain most of the phosphorylation events, a great level of confidence can be reached to conclude that that pathway is activated. Following this logic, we sorted the 23 signaling pathways according to the number of identified phosphorylation sites known to be catalyzed by their kinase members, which identified the top five pathways as PI3K-

Akt, growth hormone synthesis, ErbB, mTOR, and EGFR tyrosine kinase inhibitor resistance. Considering that our above results did not support an important role of extrinsic factors (such as cytokines and growth hormones) in JAK1/2 activation and the fact that EGFR tyrosine kinase inhibitor resistance was not relevant to our study, we dedicated our efforts to the other three signaling pathways. Notably, these pathways are intimately interconnected with each other in cancer, as ErbB signaling feeds into PI3K-Akt[32] and mTOR is a major downstream module of PI3K-Akt[27]. Importantly, our RNA-seq and phosphoproteomic analysis converged on the PI3K-Akt and mTOR pathways, highlighting their essential roles in our system. Of note, the JAK-STAT pathway was not identified by our phosphoproteomic and RNA-seq analyses; this is likely due to the preferential enrichment of peptides with serine and/or threonine phosphorylation by the TiO2-based sample preparation for phosphoproteomics, the fact that JAK-STAT proteins are primarily activated by tyrosine phosphorylation, very low basal levels of p-STAT3/1, and the dependence of these omics analysis on protein abundance. To directly test if the PI3K-Akt-mTOR axis is activated in IFNγR1^KO melanoma cells, we analyzed p-AKT and p-4E-BP1, functional readouts of mTOR action, and found both were increased (Fig. 5d). Given the co-activation of JAK1/2 and mTOR in IFNγR1^KO cells

and a recent study showing a positive mutual regulatory relationship between them in colorectal tumor cells[33], we assessed how they interact and regulate one another in melanoma. First, we took a pharmacological approach by treating cells with rapamycin (Rapa), a well-established inhibitor for mTOR and found that Rapa profoundly suppressed p-JAK2 (Fig. 5e); conversely, inhibition of JAK1/2 with Ruxo did not change p-4E-BP1 in IFNγR1[KO] cells (Supplementary Fig. 5a), placing mTOR upstream of JAK1/2. To directly assess the role of mTOR in JAK1/2 activation, we knocked down mTOR using shRNAs (mTOR[KD]) (Fig. 5f). Similar to mTOR inhibition by Rapa, mTOR[KD] also significantly reduced p-JAK1/2 and at least partially rescued JAK1/2 activation in IFNγR1[KO] cells (Fig. 5f). Collectively, these results establish that augmentation of mTOR pathway is a major upstream regulator of JAK1/2 activation in melanoma cells lacking functional IFN-γ signaling.

To establish the clinical relevance of our findings, we rationalized that IFNGR1[Low] SKCMs with impaired IFN-γ signaling and patient melanomas resistant to ICBs would house activated mTOR and JAK1/2 to some extent. Because phosphorylation data of JAK1/2 and mTOR were not available in the TCGA database and in the published database of melanoma patients treated with ICB (GSE78220)[34], precluding a direct examination of their activation, as an alternative approach, we constructed a list of genes that were reported to be direct downstream targets of mTOR and JAK1/2 in various tumor types, including bladder cancer, breast cancer, liver cancer, lymphoma, and chondrosarcoma. These genes encompass tumor promoter genes (ENO1, FASN, FKBP4, ODC1, JUNB, and VEGFA)[35–42] and tumor suppressor gene (GADD45A)[43]. Notably, activation of mTOR and/or JAK-STAT leads to upregulation of tumor promoter genes (ENO1: α-Enolase, an important glycolytic enzyme; FASN: fatty acid synthase, a major enzyme for de novo fatty acids synthesis; FKBP4: FK506-binding protein 4, an HSP90-associated co-chaperone; ODC1: ornithine decarboxylase, the first biosynthetic enzyme of the polyamine pathway; JUNB: a key member in the activator protein (AP-1) family with an important role in cell cycle progression; VEGFA: vascular endothelial growth factor-A, a key regulator of angiogenesis) but downregulation of GADD45A (the founding member of the growth arrest and DNA damage-inducible 45 families with important function in promoting cell cycle arrest and apoptosis), consistent with their prominent roles in tumor formation[27,44]. To specifically assess their expression in human melanomas, we deconvoluted the TCGA and GSE78220 databases derived from bulk tumor samples, as described above. While understandably not all the genes showed significant changes in melanoma, we did observe upregulation of ENO1, FASN, and FKBP4, as well as downregulation of GADD45A in IFNGR1[Low] SKCMs (Supplementary Fig. 5b); on the other hand, patient melanomas resistant to anti-PD-1 exhibited significant increases of ENO1, FKBP4, ODC1, and VEGFA (Supplementary Fig. 5c). The other genes exhibited expected increases/decrease, which did not reach statistical significance (Supplementary Fig. 5b, c). In spite of the differences in affected genes between the TCGA and GSE78220 databases, two genes (ENO1 and FKBP4) were consistently upregulated in both IFNGR1[Low] SKCMs and ICB non-responders, suggesting that they may be more sensitive to attenuation of IFN-γ signaling and ICB resistance. Taken together, our RNA-seq, phosphoproteomic analysis, bioinformatic analysis, as well as pharmacological and genetic modulations of the mTOR pathway establish that malfunction of IFN-γ signaling engages the mTOR-JAK1/2 axis in melanoma cells, which may represent an attractive target for therapeutic interventions to bypassing ICB resistance in melanomas lacking functional IFN-γ signaling.

### Selective suppression of IFNγR1[KO] melanomas by JAK inhibition

To test this, we employed Ruxo, an FDA-approved JAK1/2 inhibitor for myeloproliferative neoplasms (MPN), which is also being tested preclinically[45,46] and clinically in solid tumors[47], as well as in overcoming chemotherapy resistance[48,49]. However, its utility in ICB

resistance has not been explored. To this end, we treated B6 mice bearing scrambled control and IFNγR1[KO] melanomas with Ruxo. Whereas Ruxo did not result in growth suppression of scrambled control melanoma (Fig. 6a), it potently inhibited IFNγR1[KO] melanoma growth (Fig. 6b, c), highlighting a selective suppressive effect of Ruxo in the latter. Given that JAK1/2 were activated in IFNγR1[KO] cells at the baseline (Fig. 3), we asked if IFNγR1[KO] cells were more sensitive to Ruxo-induced cell killing. To this end, we first titrated out effective doses of Ruxo at suppressing JAK1/2 in scrambled control and IFNγR1[KO] cells, based on suppression of p-STAT1/3 derived from a brief stimulation of IFN-α (Note: this was necessary for a ready detection of p-STAT1/3, given the low basal level of p-STAT1/3 in these cells). As shown in Supplementary Fig. 6a, Ruxo already showed significant suppression of p-STAT1/3 at 10 nM and at 1 μM, completely blocked induced p-STAT1/3 by IFN-α. However, no appreciable killing of scrambled control and IFNγR1[KO] cells by Ruxo (10 nM–1 μM) was observed (Supplementary Fig. 6b), neither did it cause differential suppression of colony formation between these two cell types in a 7-day colony forming assay (Supplementary Fig. 6c). These data indicate that the selective suppression of IFNγR1[KO] melanoma by Ruxo is unlikely a result of the preferential killing of IFNγR1[KO] cells by Ruxo.

Next, we wondered if Ruxo treatment of IFNγR1[KO] melanoma could render TILs more functional. To this end, single-cell suspensions prepared from untreated and Ruxo-treated IFNγR1[KO] melanomas were analyzed. In line with the fact that Ruxo is a well-established JAK1/2 inhibitor, we observed the expected suppression of p-JAK2 and p-STAT3 in tumor (CD45[−]) cells by Ruxo (Supplementary Fig. 6d). Interestingly, Ruxo resulted in a pronounced reduction of T$_{reg}$ in CD4[+] TILs (Fig. 6d) and a milder but still significant reduction in CD4[+] splenocytes (Supplementary Fig. 6e), consistent with previously reported Ruxo suppression of T$_{reg}$ in humans[50] and mice[51]. Moreover, Ruxo increased TNF, IFN-γ, perforin, and IL-2 production by CD4[+] TILs (Fig. 6e), essential effector molecules in anti-tumor immunity; similar increases of IFN-γ (Supplementary Fig. 6f), perforin (Supplementary Fig. 6g), and GzmB (Supplementary Fig. 6h) were also noticed in CD8[+] TILs. Intrigued by these prominent in vivo Ruxo effects on TILs, we asked if Ruxo could directly reprogram TILs in vitro. To this end, TILs isolated from untreated melanomas were cultured with 100 U/mL IL-2, ±1 μM Ruxo (a concentration with potent suppression of p-STAT1/3 in vitro) for 3 days and then analyzed for FoxP3 expression (Fig. 6f) and production of IFN-γ/TNF (Fig. 6g). Although not as striking as the in vivo effects, this in vitro Ruxo regimen nevertheless reduced FoxP3 expression and enhanced effector function of TILs. Considering the reported on-target suppressive effects of Ruxo on MPN-associated splenomegaly that could ensue potential toxicity on mature T cells[52], we assessed the abundance of CD4[+] and CD8[+] T cells in the spleens and did not observe overt reduction (Supplementary Fig. 6i), suggesting negligible toxicity from this short-term Ruxo therapy. Because our results revealed minimal direct killing of tumor cells and substantial modulation of TILs by Ruxo, we posit that Ruxo relies on TILs to mediate its efficacy.

### T cells and host TNF signaling control Ruxo efficacy

To directly assess the importance of T cells in Ruxo therapy, we treated IFNγR1[KO] melanoma-bearing mice with anti-CD4 and anti-CD8 neutralizing antibodies prior to and during Ruxo therapy. Strikingly, deletion of either CD4[+] or CD8[+] T cells completely abolished Ruxo efficacy (Fig. 7a), supporting a pivotal role of T cells in orchestrating therapeutic effects of Ruxo. Next, we wanted to delineate the molecular mechanism(s) underscoring Ruxo efficacy. To this end, we focused on TNF for the following considerations: (1) TNF has long been regarded as an important effector molecule in mediating tumor necrosis[53] and has been previously shown to be important in anti-tumor immune responses[54]. (2) TNF has been reported to suppress T$_{reg}$ in both mouse and human systems[55,56], which coincides with the

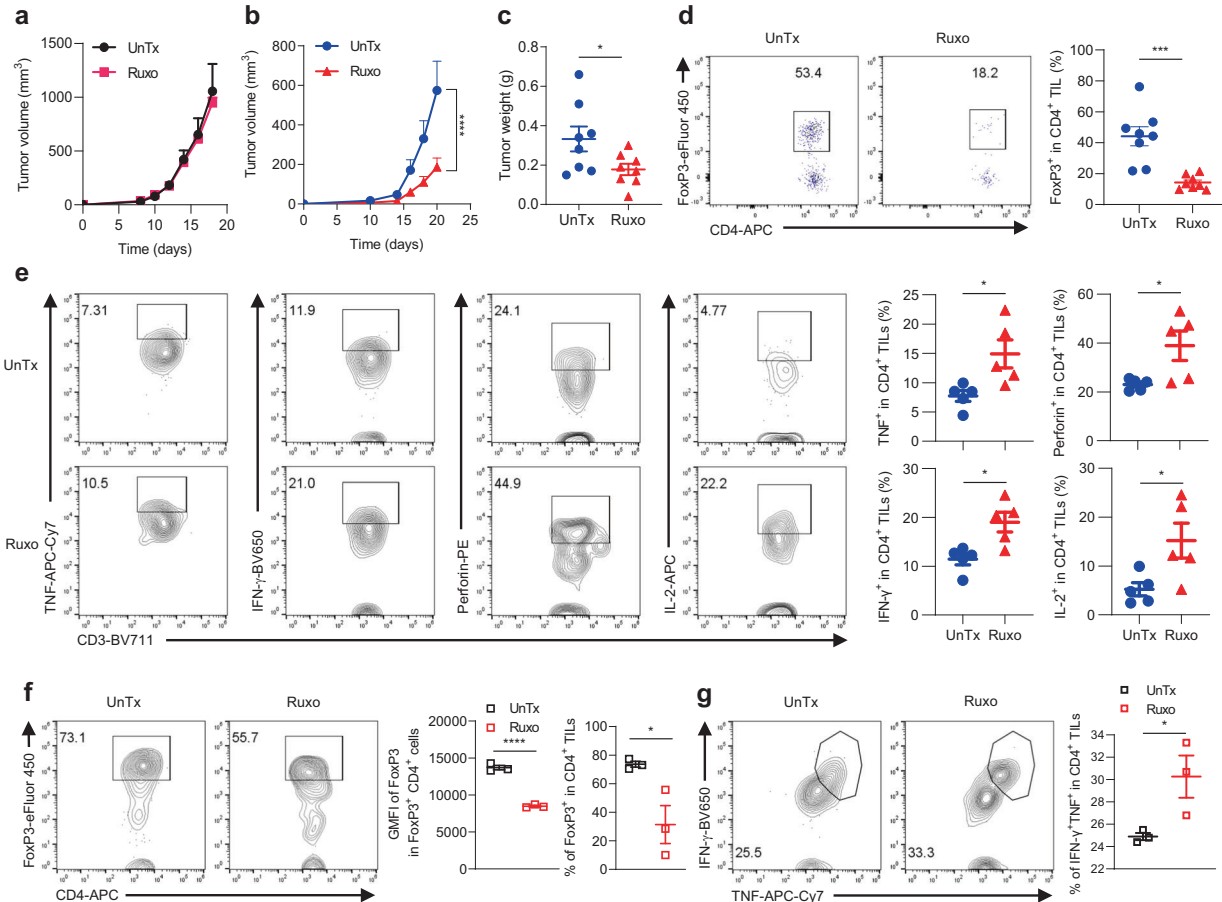

**Fig. 6 | Ruxo suppresses IFNγR1$^{KO}$ but not scrambled control melanomas.**
**a** Growth of scrambled control melanomas in B6 mice treated with vehicle (UnTx, $n = 5$) or with Ruxo (90 mg/kg by oral gavage twice daily) ($n = 5$) for 10 days. **b–e** B6 mice bearing IFNγR1$^{KO}$ melanoma were treated as in **a**. **b** Tumor growth: $n = 8$ per group; ****$p = 0.0002$ by two-way ANOVA with Šídák's multiple comparisons test. **c** Tumor weights at euthanization ($n = 8$ per group; *$p = 0.0422$). Isolated TILs from these mice were analyzed for frequency of FoxP3$^+$ T$_{reg}$ (**d**) ($n = 8$ per group; ****$p = 0.0003$) and cytokine production of TNF, IFN-γ, Perforin, and IL-2 in CD4$^+$

TILs (**e**) ($n = 5$ per group; *$p = 0.0233$ for TNF; *$p = 0.011$ for IFN-γ; *$p = 0.0311$ for Perforin; *$p = 0.0319$ for IL-2) after a brief stimulation with PMA and ionomycin. **f, g** Isolated TILs were cultured with 100 U/mL IL-2, ±1 μM Ruxo, for 3 days, to analyze FoxP3$^+$ T$_{reg}$ (**f**) ($n = 3$ per group; *$p = 0.0346$; ****$p = 0.00009$) and IFN-γ/TNF production (**g**) ($n = 3$ per group; *$p = 0.0488$) in CD4$^+$ TILs after a brief stimulation with PMA and ionomycin by flow cytometry (FACS strategy 3). A two-sided Student's $t$-test was used in **c–g** for statistical analyses. The scatter plots and line graphs depict means ± SEM. Source data are provided in the Source Data file.

prominent effect of Ruxo therapy (Fig. 6d and S6e), implying an intricate connection between Ruxo and TNF. (3) Both Ruxo and anti-CTLA-4 induced prominent production of TNF by TILs (Figs. 2c, S2a, and Fig. 6e). Because Ruxo was systemically administered in our study, we further assessed if Ruxo impacted TNF production by other immune cells such as intratumoral CD8$^+$ T cells (Supplementary Fig. 7a), dendritic cells (DCs: CD11c$^+$MHC-II$^+$, Supplementary Fig. 7b), and macrophages (CD11b$^+$F4/80$^+$, Supplementary Fig. 7c). Interestingly, no increase of TNF production by these immune cells was induced by Ruxo, suggesting a selective promotion of TNF production by Ruxo in CD4$^+$ TILs. Despite these seemingly dispensable effects of Ruxo on TNF production in these immune cells, they (in particular, CD8$^+$ TILs and macrophages, and likely, other immune cells) still produce an abundant amount of TNF, highly comparable to that of CD4$^+$ TILs (Fig. 6e), which can contribute to the overall T cell-dependent anti-tumor responses elicited by Ruxo therapy. To directly examine how the host TNF signaling affects Ruxo efficacy, we inoculated TNF$^{-/-}$ mice lacking TNF in host cells, including immune cells (T cells, myeloid cells, etc.), with IFNγR1$^{KO}$ melanoma cells, followed by Ruxo treatment. In contrast to the significant suppression of IFNγR1$^{KO}$ melanomas by Ruxo in B6 mice (Fig. 6b), Ruxo was unable to suppress IFNγR1$^{KO}$ melanoma in TNF$^{-/-}$ mice (actually, reversed) (Fig. 7b, c), highlighting a crucial role of host TNF signaling in this process. To

assess whether TNF deficiency abrogates Ruxo modulatory effects on TILs, we analyzed TILs from TNF$^{-/-}$ mice treated with Ruxo and did not observe Ruxo-driven depletion of T$_{reg}$ (Fig. 7d). Also, there was no increase of IFN-γ production by CD4$^+$ TILs (Fig. 7e) and CD8$^+$ TILs (Supplementary Fig. 7d). Similar findings were noticed for IL-2 production by CD4$^+$ (Fig. 7f) and CD8$^+$ TILs (Supplementary Fig. 7e). Considering the potentially detrimental effects from chronic TNF deficiency in TNF$^{-/-}$ mice, we took a complementary approach by temporarily blocking TNF with in vivo anti-TNF neutralizing antibodies. We treated mice before tumor inoculation and throughout the duration of Ruxo therapy. As shown in Supplementary Fig. 7f, like TNF$^{-/-}$ mice, in vivo neutralization of TNF also largely abolished the therapeutic effects of Ruxo. Lastly, considering the well-recognized role of TNF in inducing tumor necrosis, we determined if TNF could induce greater killing of IFNγR1$^{KO}$ melanoma cells as an additional underlying mechanism, in addition to the aforementioned immuno-modulatory effects. To this end, both scrambled control and IFNγR1$^{KO}$ cells were treated with TNF in vitro. Surprisingly, no obvious killing was seen, even when TNF was used at a supraphysiologically high dose (10,000 U/mL) (Supplementary Fig. 7g), suggesting that direct killing of tumor cells by TNF may not be important for Ruxo efficacy. In sum, these results indicate that Ruxo selectively suppresses the growth of IFNγR1$^{KO}$ melanoma in a T cell and TNF-dependent manner.

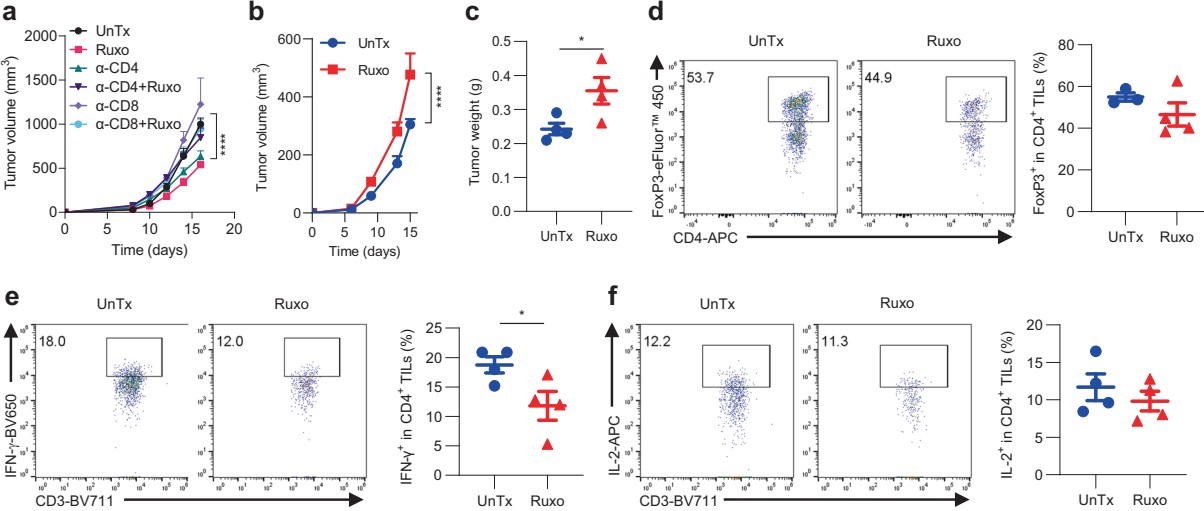

**Fig. 7 | Ruxo-induced suppression of IFNγR1^KO melanomas relies on T cells and host TNF. a** Growth of IFNγR1^KO melanomas in B6 mice treated with Ruxo, ±neutralizing antibodies against CD4^+ (α-CD4) or CD8^+ (α-CD8) T cells ($n = 5$ for UnTx, $n = 5$ for Ruxo, $n = 10$ for α-CD4, $n = 10$ for α-CD4+Ruxo, $n = 9$ for α-CD8, $n = 5$ for α-CD8+Ruxo). ****$p = 0.00007$ by two-way ANOVA with Tukey's multiple comparisons test (with adjustment). **b**–**f** TNF^−/− mice bearing IFNγR1^KO melanoma were treated with vehicle (UnTx, $n = 4$) or Ruxo ($n = 4$) (90 mg/kg by oral gavage twice daily) for 10 days. **b** Tumor growth (***$p = 0.0007$ by two-way ANOVA with Šídák's multiple comparisons test with adjustment). **c** Tumor weights at euthanization (*$p = 0.0391$) were shown. Isolated CD4^+ TILs from these mice were analyzed for FoxP3^+ T_reg frequencies (**d**) and production of IFN-γ (**e**) (*$p = 0.0473$) and IL-2 (**f**) after a brief PMA and ionomycin stimulation by flow cytometry (FACS strategy 3). A two-sided Student's $t$-test was used in **c**–**f** for statistical analyses. Representative results from two independent experiments are shown as means ± SEM in the scatter plots and line graphs. Source data are provided in the Source Data file.

## Discussion

Paradigm-shifting ICBs have brought great promises to patients with advanced melanoma, a tumor type that had been largely incurable until the approval of anti-CTLA-4 in 2011. However, therapeutic resistance to ICBs is common[8] and the loss of IFN-γ signaling in melanoma cells has been reported to be a major mechanism of resistance[9–14]. Given this key information, little is known about why this resistance occurs and how to overcome it. Here, we identify that melanomas defective of IFN-γ signaling are not only resistant to IFN-γ-induced cell death but also have reduced infiltration of CD8^+ T cells and lack of anti-CTLA-4 induced functional rejuvenation of TILs, posing a dual resistance to ICBs. Surprisingly, IFNγR1^KO melanomas harbor an aberrantly active mTOR-JAK1/2 axis, which, when targeted with an FDA-approved JAK1/2 inhibitor Ruxo, results in potent and selective suppression of IFNγR1^KO but not scrambled control melanomas, in a T cell and host TNF-dependent fashion. Moreover, human melanomas with attenuated IFN-γ signaling or ICB resistance exhibit reduced expression of T cell signature genes and alteration of target genes downstream of mTOR and JAK1/2 pathways, suggestive of their activation. Our results herein establish an important role of tumor IFN-γ signaling in modulating TILs and manifest a potential "targeted" therapy for ICB-resistant IFNγR1^KO melanomas.

Tumors lacking functional IFN-γ signaling have been shown to evade endogenous immunosurveillance[57–59] and anti-tumor immunity elicited by ICBs[9,10]. However, it is unknown whether tumor-intrinsic IFN-γ signaling modulates TILs. On one hand, IFN-γ, by upregulating MHC molecules and activating tumor antigen processing and presentation machinery[60–64], promotes anti-tumor immunity; on the other hand, it can also suppress anti-tumor immunity by inducing various regulatory mechanisms such as PD-L1 upregulation in stromal and tumor cells[65]. We observed a pronounced reduction of both MHC molecules and PD-L1 in IFNγR1^KO melanoma, albeit the former being more pronounced. Our study corroborates an early pioneering study by Bob Schreiber and colleagues, which demonstrated that IFNγR1 truncation in methA fibrosarcoma decreased tumor immunogenicity and responsiveness to LPS therapy[59]. Although our results suggest that

lack of inducible PD-L1 upregulation in IFNγR1^KO melanomas has a seemingly nonessential role in promoting TILs, this is likely a context-dependent finding, as incongruous results have been reported for the importance of tumor PD-L1 in anti-tumor immunity[66–68]. Given these findings of reduced T cell infiltration and function in IFNγR1^KO melanoma, it would be interesting to delineate the specific molecular and biochemical mechanisms underlying the immunomodulation of TILs by tumor IFN-γ signaling in the future. For example, what is the role of MHC downregulation in this process? How would tumor cell-intrinsic IFN-γ signaling regulate stemness, survival, and metabolic fitness of tumor cells, as these features have been associated with therapeutic resistance[45] and suppression of TILs' function[69]? To this end, a recent study showed that melanoma cells defective of IFN-γ signaling outgrew wild-type tumor cells when treated with anti-PD-1[70], indicating a survival advantage of IFNγR1^KO cells.

We identified a JAK1/2-centered network of constitutively active PTKs in IFNγR1^KO melanomas, which offers a "personalized" therapeutic target that can be harnessed to treat these ICB-resistant melanomas. Indeed, short-term Ruxo therapy selectively suppressed IFNγR1^KO melanomas, coupled with improved TILs' effector function and reduced frequency of intratumoral T_reg. Our results established an essential role of T cells and host TNF signaling in governing Ruxo efficacy. Although we observed that Ruxo selectively promoted TNF production by CD4^+ TILs but not by CD8^+ TILs and myeloid cells (i.e., macrophages and DCs), it is noteworthy to mention that those immune cells (esp., CD8^+ TILs and macrophages) produced abundant and comparable amount of TNF to that of CD4^+ TILs (if not higher), which can in turn act on TILs, in an autocrine or paracrine manner, to mediate therapeutic effects of Ruxo. Additionally, other immune cells such as γδ T cells, iNKT, NK cells, and innate lymphoid cells (ILCs) can also produce an ample amount of TNF that can be regulated by Ruxo. Additional mechanistic studies using mice with selective deletion of TNF in different immune cell populations (e.g., CD4^+ T cells, CD8^+ T cells, DCs, macrophages, and other immune cells) are needed to explicitly pinpoint the major cellular sources of TNF that underscore Ruxo efficacy. Importantly, Ruxo has been utilized preclinically to treat

solid tumors, with promising effects reported in ovarian cancer by suppressing stemness[45], in aggressive carcinoma by antagonizing TGF-β-induced production of leukemia inhibitory factor[46], and in KRAS-driven lung adenocarcinoma by decreasing tumor-promoting chemokines, cytokines, as well as immunosuppressive myeloid-derived suppressor cells[71]. Here, we report that Ruxo can be also utilized to overcome ICB resistance derived from tumor loss of IFN-γ signaling. Currently, Ruxo is being clinically tested in patients with advanced solid tumors (NCT02646748), non-small cell lung cancer (NCT02917993), and triple-negative breast cancer (NCT02876302)[47]. Our results justify further testing of Ruxo in patients with advanced melanoma that are resistant to ICBs, which accounts for ~75% of all patients[9]. Although our short-term Ruxo therapy was effective and did not incite overt immunosuppressive toxicity, we argue that it likely needs to be combined with other therapeutic modalities to achieve a long-term cure. To this end, preclinical studies have shown that JAKi can improve the therapeutic efficacy of radiotherapy[72–75], and when rationally combined with other chemotherapies or oncolytic virus immunotherapy, induce synergistic effects in different types of cancer[76–78].

Interestingly, a similar counterintuitive reactivation of the JAK-STAT pathway was previously identified in MPN cells that were chronically treated with JAK2 inhibitor[49]. Perhaps, long-term JAK inhibition and the chronic functional deficiency (as in IFNγR1$^{KO}$ melanoma) would engage other mechanisms to reactivate this essential pathway to sustain crucial functions such as cell division and differentiation[23]. We show here that the augmented mTOR pathway represents such a key compensatory mechanism, resulting in JAK1/2 activation in IFNγR1$^{KO}$ melanoma. However, how IFNγR1$^{KO}$ activates the PI3K-Akt-mTOR axis remains to be delineated. In a patient with myelodysplastic syndrome, the constitutively active fusion protein TEL-Syk is associated with activated PI3K-AKT[79]. And, ectopic knock-in of TEL-Syk or overexpression of Syk in various lymphoma cells[80] directly leads to activation of mTOR. Although our kinomic studies revealed active Syk, unfortunately, additional WB analyses showed that p-Syk was extremely low and did not show significant differences between scrambled control and IFNγR1$^{KO}$ cells. Future studies with genetic knockdown/knockout of Syk may be worth pursuing to directly pinpoint its involvement in mTOR activation. In addition, our phosphoproteomic studies identified activation of ErbB signaling as a top hit in IFNγR1$^{KO}$ melanoma, which is known to feed signals into the PI3K-Akt-mTOR pathway[32] and may represent a potential underlying mechanism of mTOR activation. As we recently described[23], as principal gatekeepers of various cellular signaling pathways, JAK1/2 are delicately regulated at different levels, including post-translational modifications, inhibitory function of the pseudokinase domain, as well as many regulators such as phosphatases, Protein Inhibitors of Activated STAT (PIAS) that inhibit STAT-DNA binding, and suppressor of cytokine signaling (SOCS)[81]. It would be interesting to investigate how the IFNγR1$^{KO}$-mTOR axis affects these regulatory mechanisms in the future, especially the activity of protein tyrosine phosphatases (PTPs) to mediate activation of JAK1/2.

In summary, we demonstrate that ICB-resistant melanomas lacking IFN-γ signaling have reduced infiltration and effector function of TILs but exhibit an aberrantly active mTOR-JAK1/2 axis. Inhibiting activated JAK1/2 with Ruxo induces selective suppression of IFNγR1$^{KO}$ melanomas, providing a "targeted" therapy to treat these ICB-resistant melanomas. Ruxo relies on T cells and host TNF signaling but not direct killing of tumor cells to exert its selective efficacy. Since Ruxo is clinically approved to treat MPN and is actively being tested preclinically and clinically in solid tumors[47], our findings lay a solid foundation for additional clinical testing of Ruxo in patients with advanced melanoma resistant to ICBs, which can be repurposed to overcome ICB resistance, a pressing unmet medical need.

## Methods

### Mice and cell lines

Seven-week-old C57BL/6 (Stock No: 000664), Rag-1$^{-/-}$ (Stock No: 002216), and TNF$^{-/-}$ (Stock No: 005540) mice were purchased from The Jackson Laboratory (Bar Harbor, ME) and housed in specific pathogen-free conditions in the animal facility of The University of Alabama at Birmingham (UAB) under 12 h/12 h light/dark cycle, ambient room temperature (22 °C) with 40–70% humidity. The animal protocol (APN-21945) was approved by Institutional Animal Care and Use Committee at UAB. All tumor-bearing mice were humanely euthanized prior to their tumors reaching the maximally allowed tumor size (20 mm in diameter) in our animal protocol. The B16-BL6 murine melanoma cells were kindly provided by Dr. I. Fidler at MD Anderson Cancer Center and cultured with MEM supplemented with 10% FBS, 2 mM L-glutamine, 1 mM sodium pyruvate, 1% nonessential amino acids, 1% vitamin, 100 units/mL of penicillin and 100 μg/mL of streptomycin (all from Invitrogen) in a humidified 37 °C incubator with 5% CO$_2$. B16-BL6 IFNγR1$^{KD}$ and scrambled control cells were similarly maintained and used as we previously described[9]. All cells were regularly tested using the MycoAlert detection kit (Lonza, LT07-118) and kept free of mycoplasma.

### Generation of genetically engineered cell lines

Gene knockout cell lines were generated using CRISPR-Cas9 technology, as we previously described in ref. 82. Briefly, single guide RNA sequences (sgRNAs) were inserted into the lentiCRISPR v2 plasmid (Addgene, #52961). Lentiviruses were packaged by co-transfecting 293 T cells with lentiCRISPR v2, pMD2.G (Addgene, #12259), and psPAX2 (Addgene, #12260). B16-BL6 cells were then transduced with lentiviruses containing scramble sgRNAs (5′-GCACTACCAGAGCTAACTCA-3′, targeting GFP) or sgRNAs against genes of interest. Cells were then selected with 2 μg/mL of puromycin and then seeded on 96-well-plates at ~1 cell per well. The grown single clones were then screened based on PD-L1 expression after IFN-γ and IFN-α stimulation for IFNγR1$^{KO}$ and IFNαR1$^{KO}$, respectively, with further confirmation of their IFNγR1 and IFNαR1 expression by flow cytometry. Used sgRNAs against mouse *Ifngr1* were sgRNA #2 (5′-TGGAGCTTTGACGAGCACTG-3′) and sgRNA #5 (5′-AGCTGGCAGGATGATTCTGC-3′). Used sgRNAs against mouse *Ifnar1* were sgRNA #1 (5′-TCAGTTACACCATACGAATC-3′) and sgRNA #2 (5′-GCTTCTAAACGTACTTCTGG-3′). For mTOR knockdown, lentiviruses containing shRNAs against mouse mTOR or scramble shRNA were purchased from Santa Cruz Biotechnology (#sc-35410-V). Transduction of scrambled control or IFNγR1$^{KO}$ B16-BL6 cells were performed following the manufacturer's instructions. Briefly, cells were seeded to 6-well-plate and cultured until ~70% confluency. Ten microliters of scramble or shmTOR lentivirus were added to the medium containing 8 μg/mL of polybrene from Santa Cruz Biotechnology (#sc-134220). Forty-eight hours later, cells were transferred to a 10-cm plate and selected with 2 μg/mL puromycin until no further cell death was observed with puromycin selection. Successfully transduced cells are maintained in a medium containing 1 μg/mL puromycin. mTOR knockdown was confirmed by WB. For IFNγR1 restoration, we subcloned mouse *Ifngr1* cDNA to pLenti CMV GFP Puro between BamH I and Sal I restriction enzyme sites. Lentiviruses were packaged by co-transfection with pMD2.G and psPAX2. Scrambled control and IFNγR1$^{KO}$ cells were transduced with lentiviruses, selected under 2 μg/mL puromycin, and maintained in medium containing 1 μg/mL puromycin. In some experiments, scrambled control and IFNγR1$^{KO}$ B16-BL6 cells were seeded on a six-well-plate, left untreated, or treated with 10 and 50 μg/mL of anti-IL-6 (Bio X Cell, clone MP5-20F3, #BE0046) or anti-IL-6R (Bio X Cell, clone 15A7, #BE0047,) antibodies for 48 h, and then lysed for WB analysis of phospho-JAK2 (see WB section below). To prove effective blocking with anti-IL-6 and anti-IL-6R, we treated B16-BL6 cells with IL-6 (100 ng/mL; Biolegend, #575702) in the presence or absence of 10 μg/mL anti-IL-6/IL-6R antibodies;

harvested cell lysates were analyzed for phospho-STAT3 (see WB section below).

## In vivo tumor inoculation and treatment

Seven-week-old C57BL/6 or Rag-1$^{-/-}$ mice were shaved and inoculated in the right flanks with $1.25 \times 10^5$ of B16-BL6 cells intradermally on day 0. Mice were left untreated or treated with anti-CTLA-4 (Bio X Cell, clone 9H10, #BE0131) intraperitoneally (i.p.) on days 3, 6, and 9 with 200, 100, and 100 μg per mouse, concurrently with vaccination using GVAX (GM-CSF-expressing B16-BL6 cells irradiated for 150 Gy), as we previously reported in ref. 9. C57BL/6 and TNF$^{-/-}$ mice bearing palpable melanoma were treated with Ruxolitinib (LC Laboratories, #R-6600) by oral gavage (reconstituted evenly in ORA-Plus Suspending Vehicle), twice daily at 90 mg/kg for 10 days. In vivo TNF blocking (Bio X Cell, clone XT3.11, #BE0058) was initiated 1 day before tumor inoculation at a dose of 250 μg per mouse by i.p. and repeated every three days until mice were euthanized. In vivo neutralizing antibodies against CD4 (Bio X Cell, clone GK1.5, #BE0003-1) and CD8 (Bio X Cell, clone 2.43, #BE0061) was given at a dose of 250 μg per mouse by i.p. 1 day prior to tumor inoculation and on days 1, 3, and 10 post tumor inoculation. Tumors were measured by caliper every other day starting from day 6 and tumor volumes (mm$^3$) were calculated using the formula ($0.52 \times$ length $\times$ width$^2$). The tumor-bearing mice were sacrificed when the tumor reached 20 mm in diameter. Tumors and spleens were collected at indicated times, and tumor weights were recorded.

## TILs isolation and splenocyte preparation

Tumors were collected in ice-cold RPMI 1640 containing 2% FBS and minced into fine pieces, followed by digestion with 400 U/mL collagenase D (Worthington Biochemical Corporation, #LS004186) and 20 μg/mL DNase I (Sigma, #10104159001) at 37 °C for 40 min with periodic shaking. EDTA (Sigma, #1233508) was then added to the final concentration of 10 mM to stop digestion. Cell suspensions were filtered through 70 μM cell strainers, and TILs were obtained by collecting the cells in the interphase after Ficoll (MP Biomedicals, #091692254). Spleens were collected in ice-cold HBSS containing 2% FBS to prepare single-cell suspensions after lysis of red blood cells and filtering with 70 μM nylon mesh. Both TILs and splenocytes were resuspended in complete Click's culture medium (Irvine Scientific, #9195-500 mL) for flow cytometric analyses. In some experiments, isolated TILs were cultured with 100 U/mL IL-2, with or without 1 μM Ruxo for 3 days and analyzed for FoxP3 expression and production of IFN-γ/TNF by flow cytometry, as described below.

## Flow cytometric analysis

Surface staining of TILs and splenocytes was done in DPBS containing 2% BSA for 30 min on ice. To analyze FoxP3, following surface staining, cells were fixed using the Foxp3/Transcription Factor Staining Buffer Set (Invitrogen, #00-5523-00) and stained for FoxP3, according to the manufacturer's instructions. To detect intracellular cytokines, cells were briefly stimulated for 4–5 h with Phorbol 12-myristate 13-acetate (PMA, final concentration: 50 ng/mL; Sigma, #P8139-5MG) plus ionomycin (final concentration: 1 μM; Sigma, #I0634-1MG) in the presence of monensin (BD Biosciences, #51-2092KZ) (for the last 2 h). Stimulated cells were stained with surface markers, fixed using the BD Cytofix/Cytoperm Plus Fixation/Permeabilization Kit (BD Biosciences, #554715), and stained for cytokines according to the manufacturer's instructions. Antibodies used include Aqua fixation LIVE/DEAD™ Fixable Aqua Dead Cell Stain Kit (1:200, Thermo Fisher, #L34966), CD4-BV421 (1:200, clone RM4-5, BioLegend, #100544), CD8-BV786 (1:200, clone 53-6.7, BD Biosciences, #563332), CD45-PerCP-Cyanine5.5 (1:200, clone 30-F11, Thermo Fisher, #45-0451-82), CD11b-PE (1:200, clone M1/70, BioLegend, #101208), CD11c-APC (1:200, clone N418, BioLegend, #117310), F4/80-BV785 (1:200, clone BM8, BioLegend, #123141), TCRβ-APC Cy7 (1:200, clone H57-597, BioLegend, #109220),

CD3-BV711 (1:200, clone 145-2C11, BioLegend, #100349), IFNγR1-BV605 (1:200, clone GR20, BD Biosciences, #745111), IFNαR1-APC (1:200, clone MAR1-5A3, BioLegend, #127313), PD-L1-APC (1:200, clone 10 F.9G2, BioLegend, #124312), MHC I-BV650 (1:200, clone SF1-1.1, BD Biosciences, #742434), MHC II-BV785 (1:200, clone M5/114.15.2, BioLegend, #107645), FoxP3-eFluor™ 450 (1:100, clone FJK-16s, Thermo Fisher, #48-5773-82), Perforin-PE (1:100, clone S16009A, BioLegend, #154306), TNF-APC Cy7 (1:100, MP6-XT22, BioLegend, #506344), PD-1-APC (1:100, clone RMP1-30, Thermo Fisher, # 17-9981-82), CD73-BV605 (1:200, clone TY/11.8, BioLegend, #127215), Granzyme B-FITC (1:100, clone QA16A02, BioLegend, #372206), IFN-γ-BV650 (1:100, clone XMG1.2, BioLegend, #505832), IL-2-BV711 (1:100, clone JES6-5H4, BioLegend, #503837), phospho-JAK2 (Tyr 1007/Tyr 1008)-APC (1:100, clone E132, Abcam, #ab200340) and phosphor-STAT3 (Tyr705)-FITC (1:100, clone LUVNKLA, Thermo Fisher, #11-9033-42). For cell apoptosis analysis, cells treated with or without IFN-γ (100 U/mL), IFN-α (100 ng/mL), Ruxo (10–1000 nM), and TNF (100–10,000 U/mL) were washed once with DPBS and then washed again with 1× Annexin V binding buffer. Afterward, cells resuspended in the Annexin V binding buffer were stained with Annexin V (1:50, Thermo Fisher, #17-8007) and 7-AAD (1:200, Sigma, #129935) for 30 min at room temperature. For cell proliferation analysis, cells were pre-labeled with 4 μM Cell-Trace Violet (CTV, Thermo Fisher, #C34557) by incubating for 20 min with periodic mixing. After incubation, cells were washed twice with a complete culture medium to remove soluble CTV. CTV-labeled tumor cells (10,000 cells) were seeded onto a six-well plate to evaluate cell proliferation (CTV dilution) after being cultured in a hypoxic (1% O$_2$) and a normoxic (21% O$_2$) incubator for 72 h, in the absence and presence of IFN-γ (100 U/mL). All the flow cytometric data were acquired using the built-in software of the Attune NxT Flow Cytometer (Invitrogen, A24860) from Thermo Fisher. Flow cytometric data were analyzed using FlowJo (version 10.8.1).

## Western blot (WB)

Western blot was performed, as previously described in ref. 82. Briefly, 0.5 millions of scrambled and IFNγR1$^{KO}$ B16-BL6 tumor cells were seeded onto a 6-cm-plate and cultured for 24 h. Cells were washed with cold DPBS twice before lysed with M-PER buffer (Thermo Scientific, #78501) containing proteinase inhibitors cOmplete (Roche, #11836170001) and phosphatase inhibitors (Sigma, P2850, and P5726) directly on the plate. Lysates were then collected and transferred to 1.5 mL Eppendorf tubes and briefly sonicated. Protein concentration was determined by BCA quantification (Thermo Scientific, #23225). Fifty μg of total proteins were loaded onto each lane of a 10% SDS-PAGE gel; after electrophoresis, proteins on the gel were transferred to 0.22 μm of nitrocellulose membrane (Bio-Rad, #1620112) in a sponge sandwich. Membranes were then blocked with 5% of non-fat milk (Bio-Rad, #170-6404) and probed with primary antibodies overnight on a shaker in a cold room. After that, membranes were washed and incubated with HRP-conjugated secondary antibodies at room temperature for 1 h. The membranes were then incubated with Western HRP substrate (Millipore, WBLUR0500) for 2–5 min before imaging with an X-ray film. For p-STAT1/3 detection, substantially more total proteins (100 μg and above) were loaded onto each lane of the gel and membranes were exposed for a much longer time (20 min or longer) to enhance the signals. About 100 U/mL IFN-γ was added for the last 15 min for JAK-STAT signaling activation and phospho-JAK2 was detected. Cells were treated with or without 10 μM of Ruxo for 30 min or 1 h, 0–1000 nM of Ruxo for 2.5 h and 10 ng/mL of IFN-α for the last 15 min, 1 μM of Rapamycin for 3 h, or 0–100 ng/mL of IFN-α for 15 min as indicated in individual experiments. For supernatant treatment experiments, supernatants collected from ~70% confluent cultures of scrambled control and IFNγR1$^{KO}$ cells were spun down, filtered with 0.22 μm PVDF membrane, and used to treat cells for 24 h. The antibodies used for WB are: phospho-JAK1 (Tyr 1022) (1:1000, Santa Cruz

Biotechnology, polyclonal, #sc-101716), total-JAK1 (1:1000, Santa Cruz Biotechnology, clone HR-785, #sc-277), phospho-JAK2 (Tyr 1007/Tyr 1008) (1:1000, Santa Cruz Biotechnology, polyclonal, #sc-16566-R), total-JAK2 (1:1000, Santa Cruz Biotechnology, clone C-10, #sc-390539), phospho-AKT (Ser473) (1:1000, Cell Signaling Technology, polyclonal, #9271), total-AKT (1:1000, Cell Signaling Technology, polyclonal, #9272), phospho-4EBP1 (Thr37/46) (1:5000, Cell Signaling Technology, clone 236B4, #2855), phospho-STAT1 (Tyr701) (1:1000, Cell Signaling Technology, clone 58D6, #9167), total-STAT1 (1:1000, Cell Signaling Technology, polyclonal, #9172), phospho-STAT3 (Tyr705) (1:1000, Cell Signaling Technology, D3A7, #9145), total-STAT3 (1:1000, Cell Signaling Technology, 79D7, #4904), phospho-Syk (Tyr525/526) (1:1000, Cell Signaling Technology, C87C1, #2710), phospho-ZAP70 (Tyr493) (1:1000, Cell Signaling Technology, polyclonal, #2704 T), phospho-EphA3 (Tyr779) (1:1000, Cell Signaling Technology, D10H1, #8862 S), mTOR (1:1000, Cell Signaling Technology, clone 7C10, #2983), and β-actin (1:10000, Santa Cruz Biotechnology, #sc-47778 HRP). β-actin was run on the same blot with proteins of interest. Uncropped and unprocessed scans of all blots were provided in the Source Data file.

### RT-PCR
Total RNAs were extracted from scrambled control and IFNγR1$^{KO}$ cells using the RNeasy Plus Mini kit (QIAGEN, #74136). First-strand cDNAs were synthesized by SuperScript III reverse transcriptase (Invitrogen, # 11752250). Quantitative RT-PCR was performed on Bio-Rad One-step with primers synthesized by IDT. Primers used were *Irf-1* (Forward: 5′-CAGAGGAAAGAGAGAAAGTCC-3′; Reverse: 5′-CACACGGTGACAGTGCTGG-3′), *Il-6* (Forward: 5′-CTGCAAGAGACTTCCATCCAG-3′; Reverse: 5′-AGTGGTATAGACAGGTCTGTTGG-3′), *Il-6r* (Forward: 5′-GCCCAAACACCAAGTCAACT-3′; Reverse: 5′-TATAGGAAACAGCGGGTTGG-3′), IFNαR1 (Forward: 5′-CATGTGTGCTTCCCACCACT-3′; Reverse: 5′-TGGAATAGTTGCCCGAGTCC-3′). β-actin was used as the housekeeping gene (Forward: 5′- CATTGCTGACAGGATGCAGAAGG-3′; Reverse: 5′-TGCTGGAAGGTGGACAGTGAGG-3′). The gene expression level was calculated using the 2$^{-ΔΔCT}$ method.

### Colony formation assay
Three hundred scrambled control and IFNγR1$^{KO}$ B16-BL6 cells per well were seeded on six-well plates; triplicates were set up for each condition. About 100 nM or 100 nM of Ruxo or an equal volume of solvent (DMSO) were added to cells after seeding. Cells were cultured for 7 days following crystal violet staining. Stained cells were washed with DPBS and dried on filter paper for photographs.

### Kinomic analysis
Kinomic profiling was performed in the UAB Kinome Core. Scrambled control and IFNγR1$^{KO}$ B16-BL6 cells were lysed on ice as described in sample preparation for WB. Lysates were loaded at 15 μg per array. Each array had a porous 3D surface imprinted with tethered phosphorylatable targets. These 12–15 amino acid targets (as listed in the attached array layout file) were imprinted as "spots" in a 12 × 12 grid. Each one of these spots had thousands of identical peptide targets, with residues that could be phosphorylated as lysates were pumped through the porous array, with phosphorylation detected with phosphor-specific FITC conjugated antibodies. After each pumping cycle, the lysate itself was pumped behind an opaque membrane, and an image of the array was captured over multiple exposure times (10, 20, 50, 100, and 200 ms). Gridding of whole array images was done with Evolve 2 image analysis software prior to import into BioNavigator, where signals by exposure slopes were calculated, multiplied by 100, and log2 transformed to generate single values per peptide, per sample. These values were used for upstream kinase identification. Specifically, peptides with acceptable curve fit and signal were used to identify upstream kinases using BioNavigator Upkin PTK v6.0. Scores

derived from Kinexus (www.phosphonet.ca) for each phosphorylatable peptide residue (links in array layout file), with amino acid sequences with greater than 90% homology were queried. Kinases with PhosphoNET V2 scores greater than 300 and rank ordered in the top 12 were retained. Individually in vitro identified peptide targets of kinases on-chip from PamGene's proprietary database were given a rank order of 0. For each kinase (*ALK*), a difference between experimental groups (*T*; mean kinase statistic [MKS]) was calculated. The sample mean $\bar{p}_{ij}$ and variance $s_{ij}^2$ of peptide *i* in each comparative group. A significance score was based on permutations of samples and measured how much *T* depends on the experimental grouping of the samples. A specificity score was based on the permutation of peptides and measured how much τ depended on the peptide to kinase mapping.

$$\tau_{KINASE} = \frac{1}{n} \sum_{i=1}^{n=9} \frac{\bar{P}_{i_1} - \bar{P}_{i_2}}{\sqrt{s_{i_1}^2 + s_{i_2}^2}} \tag{1}$$

The combined overall or mean final score (MFS) was either specificity (if singlicate) or the sum of significance and specificity. Kinases identified were uploaded as seed nodes by UniProt ID to GeneGo MetaCore, where they were overlaid on literature annotated interactions, in an auto-expand network model where sub-networks were generated from the seed node list, expanded iteratively with preference given to objects with more connectivity to the initial seed nodes. The expansion was halted when the sub-networks intersected or when the network reached a selected size (*n* < 50 nodes). Networks were named by their most centric (interconnected) node.

### RNA-seq analysis
Scrambled control and IFNγR1$^{KO}$ B16 melanoma cells were seeded overnight as triplicates before RNA extraction. Cells were directly lysed on the plate and total RNA was extracted immediately by RNeasy Plus Mini Kit from QIAGEN, Inc. Standard RNA-seq was performed by GENEWIZ, Inc. Briefly, total RNA was enriched with Poly A selection and sequencing was performed on the Illumina platform. For RNA-seq data analysis, paired-end transcriptome sequences were mapped to the *Mus musculus* GRCm38 reference genome available on ENSEMBL using the STAR aligner (version 2.7.5a. Read counts per gene were calculated using htseq-count in the HTseq package (version 0.11.2)[83]. Then the read counts per gene were used for downstream differential gene expression analysis and pathway enrichment analysis. The analysis of differentially expressed genes (DEGs) between the scrambled control and IFNγR1$^{KO}$ samples was performed using DESeq2 (version 1.34.0)[84] in R (version 3.6.0). The Wald test was used to calculate the *p* values and log2 fold changes. Genes with an adjusted *p* value < 0.05 and absolute log2 fold change > 1 were considered as DEGs. A volcano plot was used to show all upregulated and downregulated DEGs using the ggplot2 package (version 3.3.6) (ggplot2: Elegant Graphics for Data Analysis. Springer-Verlag New York. ISBN 978-3-319-24277-4, https://ggplot2.tidyverse.org). Enriched Kyoto Encyclopedia of Genes and Genomes (KEGG) pathways[85] of the DEGs were identified by enrichr package[86] (version 3.0), a comprehensive gene set enrichment analysis tool. Significant terms of the KEGG pathways were selected with a *p* value < 0.05.

### Multiplexed phosphoproteomic analysis
Cells collected at 90–95% confluence were washed with ice-cold DPBS thrice and lysed in 8 M urea buffer. The protein concentration was measured with the Bradford method using Pierce™ Coomassie Plus Assay Reagent (Thermo Fisher, #23238). For each sample, 1 mg of protein was digested by TPCK-trypsin at the ratio of 50:1 (w/w) overnight at 37 °C. The peptide concentration was quantified using Pierce™ Quantitative Colorimetric Pierce Quantitative Colorimetric Peptide Assay Kit (Thermo Fisher, #23275). From each quantified peptide

sample, 70 μg of peptides was labeled using TMTpro™ 16plex Label Reagent Set (Thermo Fisher, #A44520) according to the manufacturer's manual. Labeled peptides were pooled (4 samples/group × 4 groups) and dried by Speedvac. Dried peptides were then dissolved in 0.1% trifluoroacetic acid (TFA), with pH values adjusted to 3.5 using 5% TFA. Phosphorylated peptides were enriched using $TiO_2$ beads as described previously[87]. Enriched phosphopeptides were then fractionated using the Pierce Reversed-Phase Peptide Fractionation Kit (Thermo Fisher, #84868). The fractions of total phosphopeptides were dried by Speedvac and purified using Millipore ZipTip with 0.6 μL C18 resin (Thermo Fisher, #ZTC18S096) according to the manufacturer's manual. Purified peptides were analyzed using the SPS-MS3 approach with the Orbitrap Fusion Lumos mass spectrometer[88]. Maxquant (version 1.6.17.0) was used to search against mouse protein databases that were downloaded from uniprot.org. Protein phosphosites were compared among groups based on corrected reporter ion intensities. The phosphoproteomic data generated in this study have been deposited in the Mass Spectrometry Interactive Virtual Environment (MassIVE) database under accession ID MSV000087796.

## Bioinformatic analysis

Gene expression data of The Cancer Genome Atlas (TCGA), Skin Cutaneous Melanoma (SKCM), and Uveal Melanoma (UVM) were downloaded from National Cancer Institute Genomics Data Commons (GDC) [https://gdc.cancer.gov/about-data/publications/pancanatlas]. The clinical information for each patient in TCGA was obtained from Genomic Data Commons (GDC) Data Portal [https://portal.gdc.cancer.gov/]. The gene expression profiles of published pretreatment melanomas undergoing anti-PD-1 therapy transcriptome data[89] were retrieved from the gene expression omnibus database (GEO) using the accession number GSE78220. The SKCM samples were grouped into IFNGR1$^{High}$ and IFNGR1$^{Low}$ groups based on the median expression of IFNGR1 expression in tumor cells of all samples. The statistical significances for gene expressions in IFNGR1$^{High}$ vs IFNGR1$^{Low}$ SKCMs, SKCMs vs UVMs, and anti-PD-1 responders vs non-responders were calculated using R with the Mann–Whitney U-test. To identify malignant cells from the TCGA and GSE78220 datasets, CIBERSORTx tool[20] was used to assess the cell type abundance from the transcriptomes of the bulk tumor tissues. Specifically, a matrix of reference gene expression signatures was provided as an input of CIBERSORTx (deconvolution), which were collectively used to estimate the proportions of melanoma cells and other stromal cells, including immune cells. The permutation was set as 1000, and the B-mode of batch correction was applied. Samples with $p$ value < 0.05 were considered successful deconvoluted samples. For the TCGA cohort, tumor-dominant samples were identified as the samples that had a relative signature score of the malignant cell (melanoma cell) >80%. For the GSE78220 cohort, tumor-dominant samples were those with a relative signature score of the malignant cell (melanoma cell) >60%.

## Statistical analysis

For animal experiments, five mice were included in each group; for in vitro studies with cells, triplicates were set up to ensure consistency and reproducibility. All experiments were repeated for two to five times. Preclinical results were expressed as mean ± SEM. Data were analyzed using a two-sided Student's $t$-test, one-way ANOVA, or two-way ANOVA after confirming their normal distribution. The log-rank test was used to analyze survival data from the preclinical studies. All analyses were performed using Prism 9.4.0 (GraphPad Software, Inc.) and $p < 0.05$ was considered statistically significant. TCGA data and GSE78220 data were expressed as boxplots, with the box depicting the first (lower) quartile, median, and the third (upper) quartile, and the lines indicating minimum score and maximum score. To assess the overall survival of patients with clinical information from TCGA, the survival time was calculated based on their vital status. The overall

survival of patients with IFNGR1$^{High}$ or IFNGR1$^{Low}$ SKCMs was estimated with Kaplan-Meier analysis and the differences between the cohorts were assessed with a log-rank test using the "Surv" function in the R package "Survival" (version 3.2.13). A $p$ value threshold of 0.05 was used to identify the significantly different survival rates between groups.

## Reporting summary

Further information on research design is available in the Nature Research Reporting Summary linked to this article.

## Data availability

The publicly available skin cutaneous melanoma and uveal melanoma TCGA data used in this study are available in National Cancer Institute Genomics Data Commons (GDC) [https://gdc.cancer.gov/about-data/publications/pancanatlas]. The publicly available gene expression profiles of published pretreatment melanomas undergoing anti-PD-1 therapy transcriptome data used in this study are available in the GEO database under accession code GSE78220. The RNA-seq data generated in this study have been deposited in the Gene Expression Omnibus (GEO) database under accession code GSE201078. The phosphoproteomic data generated in this study have been deposited in the Mass Spectrometry Interactive Virtual Environment (MassIVE) database under accession ID MSV000087796. All deposited data were publicly available. The remaining data in this study are available within the manuscript or Supplementary Information, with source data provided herein. Source data are provided with this paper.

## Code availability

The codes for analyzing TCGA, GSE78220, and GSE201078 data were deposited and publicly available in https://github.com/huang1990/IFNGR1_NC_paper[90].

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

## Acknowledgements

We would like to acknowledge other members of Shi lab and the Department of Radiation Oncology at UAB for their constructive input. We are grateful for the Startup fund from the Department of Radiation Oncology and the O'Neal Invests pre-R01 Grant from the UAB-O'Neal Comprehensive Cancer Center granted to Shi lab. This study is also funded by National Institutes of Health grants (1R21CA230475-01A1 and 1R21CA259721-01A1), the V Foundation Scholar Award (V2018-023), a DoD-Congressionally Directed Medical Research Programs grant (ME210108), a Cancer Research Institute CLIP Grant (CRI4342), an American Cancer Society Institutional Research Grant (91-022-19), and National Institute of General Medical Sciences (1R35GM138212).

## Author contributions

H.S. designed and did the experiments with cells and mice, analyzed data, and contributed to writing the manuscript; F.H. and Z.C. performed the bioinformatic analyses of the TCGA and GSE78220 datasets and contributed to writing the manuscript; X.Z. conducted the phospho-proteomic studies, analyzed data, and contributed to writing the manuscript; O.A.O., Y.L., and H.Q.T. did the experiments with cells and/or mice; J.C.A. and C.D.W. performed the kinomic studies, analyzed data, and contributed to writing the manuscript; J.F., E.S.Y., and M.B. contributed to manuscript construction and discussion; L.Z.S. and J.A.B. were responsible for the original conceptualization of this study, overall data presentation, and manuscript construction; L.Z.S. acquired funding for this study, designed experiments, supervised laboratory studies and data analyses, wrote, and edited the manuscript. All authors have met the requirements for authorship and are in consensus on the content of this publication.

## Competing interests

The authors declare no competing interests.
