## [Peer Review File · Nature Communications]

Selective suppression of melanoma lacking IFN- γ pathway by JAK inhibition depends on T cells and host TNF signalingReviewers' comments:

Reviewer #1 (Remarks to the Author): with expertise in cancer immunology/immunotherapy

This manuscript by Shen et al. examines IFN γ -sensing in human and mouse melanoma cells and finds a paradoxical increase in phosphorylated JAK1/2, which are the signaling kinases downstream of the IFN γ receptor. This is an intriguing finding, and suggests that either 1) other cytokines besides IFN γ are providing growth signals to the tumor cells or 2) tumor-intrinsic signaling pathways are altered to compensate for the lack of IFN γ R signaling. Given the central importance of IFN γ in regulating anti-tumor immunity in both mice and humans, this surprising finding of rewiring of tumor cell cytokine signaling is important and novel. Furthermore, drugs targeting members of the JAK family are approved for rheumatologic diseases and are being evaluated in clinical trials for several cancer indications.

That said, there are major logical flaws in the manuscript as currently presented. It is not at all clear to me that the authors understand the significance of their own work, given that both the title and the abstract inappropriately focus on CD8 T cells and misrepresent the central finding of the manuscript. Furthermore, the obvious question of what signaling pathways are activated in the tumor cells was only superficially addressed, with limited number of tumor cell clones evaluated, known JAK1/2 targets like phospho-STAT1 not examined, and responsiveness to type I IFN signaling not addressed. I am still left wondering whether IFN γ R1KO cells have increased reliance on a different cytokine or growth factor, whether these tumor cells are more exquisitely sensitive to TNF α . Furthermore, the authors have not entirely convinced me that the results shown are not just artifacts from generation of clonal cell lines.

Major concerns:

- 1) The title is very misleading as the authors do not examine the role of JAK inhibition in T cells. Please consider rephrasing to "IFN γ -pathway deficient melanomas are sensitive to JAK inhibition" or similar title that focuses on the primary message of the paper, which is the altered JAK signaling in IFN γ R1KO tumor cells. The one sentence summary doesn't make any sense "Tumor-intrinsic loss of IFN-g signaling confers immunotherapeutic resistance and suppresses TILs, which can be treated with by Ruxolinitib." What does "suppresses TILs" mean? Consider rephrasing to "Tumor-intrinsic loss of IFN-g signaling confers immunotherapeutic resistance which can be reversed by Ruxolinitib."
- 2) The experiments looking at tumor infiltrates in IFN γ R1KO tumor-bearing mice +/- anti-CTLA4 blockade are misleading. If a tumor is responding to immunotherapy and becoming smaller, then cytokine-producing CD8 effector T cells in the TILs will increase. If the tumor is not responding to anti-CTLA-4 due its known genetic loss of IFN γ R1 which has already been reported to confer both primary and acquired resistance, then it will be larger and have fewer activated T cells. This is not surprising or interesting.
- 3) The experiments looking at tumor infiltrates in IFN γ R1KO mice treated with Ruxo are also misleading. Systemic administration of a JAK inhibitor will broadly affect cytokine signaling in all cells, including immune cells. The authors cannot conclude that Ruxo treatment is acting on the increased phospho Jak1/2 in tumor cells as its primary mechanism of action without thoroughly addressing the role of Jak1/2 in immune cells as well. Furthermore, all of the effects seen could be due to altered sensing of cytokines other than IFN γ .
- 4) Jak1/2 are downstream of multiple cytokine receptors, including ones such as IL-6 that serve as direct tumor growth factors. The authors use conditioned media to examine whether autocrine or paracrine factors are secreted by tumor cells may account for increased Jak1/2 activity in IFN γ R1KO cells. These experiments were apparently negative, but the data shown are not convincing, and I would argue that a soluble ligand is still a likely explanation for the alterations observed. Proteomic profiling of the media and cell pellets would be more informative.
- 5) The phosphosite profiling is very interesting and should be explained more clearly to the non-specialist audience. It is unclear to me why STAT proteins, known targets of JAK1 and JAK2, were not identified by this analysis? Could the authors show pSTAT1 by immunoblotting? As this is the key transcription factor activated by IFN γ R, STAT1 is of particular importance whether it is changed or unchanged.

- 6) The data from TNF α blockade treated mice are intriguing, and suggest a clear mechanism of IFN γ R1KO tumor cells having increased sensitivity to TNF α -mediated death pathways. Please examine in vitro the relative TNF α sensitivity of control and KO cell lines.
- 7) Figure 5I: Please include Ruxo, TNF α , and Ruxo+TNF α treatment of control B16 tumors as well for a comparison to the IFN γ R1KO tumor growth curves.
- 8) Please justify the use of clones rather than populations of IFN γ R1KO cells. Single cell expanded clones can have altered growth behavior that deviates from the parental cell line merely by chance, and not necessarily related to the specific gene deleted. If clones were used, a minimum of 4 clones must be evaluated, or a single knockout clone of interest must be reconstituted with IFN γ R1 to show that phospho JAK1/2 levels are restored to baseline.

Minor concerns:

- 1) Line 113 "scrambled cells" is an odd phrase. Consider "scramble control cells" instead.
- 2) Lines 347-351: "We reason that if downregulation of MHC II on IFN γ R1KO melanoma plays a dominant role in shaping TILs, immunosuppression of TILs would ensue; conversely, if reduced expression of PD-L1 on tumor cells is important, immunostimulation of TILs would occur. The overall reduced abundance and function of TILs support downregulation of MHC but not PD-L1 is the more dominant factor in shaping TILs in the absence of tumor-intrinsic IFN- γ signaling." This is an overly simplistic explanation. Please do not refer to suppression or stimulation of TILs as a bulk entity as TILs encompass a variety of distinct cell types with distinct functions.
- 3) Figure 1A: Why are the bands pictured not at the same height across the 4 samples? Were these run on different gels? Supplemental Figure 1C should be added to main Figure 1 for comparison.
- 4) Figure 1E: Please report actual tumor measurements in cubic mm rather than fold change. If the IFN γ R1KO clone tumors grow differently from the parental line, this is important information to convey.
- 5) Figure 2 flow plots are shown as dots plots, a heatmap, and contour plots. Please pick one graphical display format to use consistently throughout.
- 6) Figure 2: Tumor size at time of harvest needs to be shown. CD8 T cells vary based on tumor size, so if the IFN γ R1KO tumors, which are unresponsive to anti-CTLA4 therapy, were larger at the time of harvest, then the results presented here are entirely expected and could be moved to supplemental.
- 7) Figure 4 and Supplemental Figure 6: It appears from the data shown that all melanoma cells are sensitive to Ruxo, not just the IFN γ R1KO cells. Please change the figure titles and accompanying text to not mislead the readers into thinking that IFN γ R1KO cells are preferentially sensitive.

Reviewer #2 (Remarks to the Author): with expertise in cancer biology, phosphoproteomics

NCOMMS-21-25916

Hongxing Shen and Lewis Shi present a manuscript entitled "JAK inhibition reprograms intratumoral T cells and bypasses tumor intrinsic immunotherapeutic resistance"

Dr. Shi pioneered the discovery of IFN gamma signaling as a mediator to immune checkpoint inhibitors. In this manuscript, as a senior author, his group reports mechanistic cues on this phenomenon, mainly consisting on an aberrant signaling axis involving JAK, which in turns regulates TILS infiltration rate and their functionality. This axis is therapeutically targetable, and thus the information contained in this manuscript could provide a basis for a proof of concept clinical trial. The described mechanisms is new and sheds light over a significant problem in oncology, and thus, it

could warrant publication in Nature Communications. However, a few major observations impede the publication of the manuscript as it present form, particularly regarding the description of the mechanisms and the proteomics approaches.

1) Mechanistic explanation. The main point here is that JAK1/2 de-regulates TNF signaling and this affects negatively to the infiltration and activation of TILs. This is well shown and I accept this part. However, at the same time, the authors defend that Ruxo has cell-autonomous effects (whole Fig 4 data). This, on itself, is counter-intuitive and unnecessary. If the effect is so strong in a cell-autonomous manner, then, what is the role of JAK in vivo? Kill the cells? Or reactivate the tumor infiltrating lymphocytes? It can not be claimed the latter if the tumor cell specific is so strong. Experiments depleting CD4 and/or CD8 cells (or their mediators IFN, TNF, IL2,... shown in Fig 5) in vivo should allow weighting whether the effect is more Ruxo-mediated cell killing or more Ruxo-mediated TIL infiltration and invigoration. If it is because of cell killing, the claim of infiltration and invigoration invalidates the whole point of the manuscript. However, an alternative hypothesis is possible, which is that the authors used in vitro 20 micromolar of Ruxo, which is something that can not be achieved in vivo in the stroma (1000-fold less, actually) and it may be killing cells independently of JAK (off target) because of ultra high dose. If this is the case (test killing non-JAK-dependent models, or JAK Kos, or test lower dose), then, the in vitro data add nothing and should be removed. If the claim is to defend that this is the reason ICB fails, authors do not need to show any cell-autonomous effect of the drug that reverses the ICB failure, they only need to show that they rescue ICB activity, and actually this would be better supported in absence of direct activity of the drug in the tumor cells (cleaner effect). I suggest clarifying this point in vivo, and formally excluding cancer cell killing by Ruxo (non-specific, or at least, not ICB failure-rescue mediated). Same for the in vitro data with the human cell line.

2) Proteomics data generation – Involvement of JAK1/2 and serine/threonine kinases in tumors with low IFN gamma R1.

a) Data for detecting tyrosine kinases aberrantly activated

a.1) The chip approach is not a widespread technique and it should be more thoroughly described. Both technically and analytically. If there are text length limitations, it should go in the material and methods. Primary, and more importantly, secondary analysis, should be thoroughly described (i.e., “how they come up with these kinases”). Chips have many issues for proteomics (sensitivity, specificity, semiquantitative at best), although less when using a limited number of targeted proteins, and should be discussed. I would be more convinced from a mass-spectrometry guided approach following a tyrosine phosphorylation enrichment step.

a.2). Since many (26) kinases were observed as enriched (and the number of tyrosine kinases is not that large) I suggest at least discarding the involvement of those with high scores, other than JAK, in supplementary figures similar to 3B, blotting other kinases. It may occur, due to the network nature of these signaling events, that there are more upstream players that, when JAK is therapeutically targeted, keep feeding the network, lacking a final positive outcome.

a.3.) The analytic approach overlying genomics and proteomics to pinpoint JAK (3a) has to be described in more detail, since most readers will not be familiar with it.

b) Data for detection of aberrantly active serine/threonine kinases

b1) Although the mass spectrometry approach looks state of the art, only the explanation for the primary analysis (i.e., peptide identification, quantitation, etc) is provided. I particularly miss information about the secondary analysis. For example:

-How were the proteins shown in Figure 3C selected? How come there are not hundreds, if not thousands, of other proteins regulated in clone 1, clone 2 or KD versus scrambled? The cut-off score for showing only those should be shown, explained and justified.

-The previous question entails with the fact that many other proteins not shown there may map to other kinases implicated in the loop. Why where they filtered out?

-The highlighted phosphopeptides in red are substrates of AKT, MTOR and p70S6K according to the authors. How was this concluded? Phospho-peptides actually have a long consensus sequence that can in fact be target of many kinases, and have higher affinity or lower affinity to a diverse number of kinases, it is not “black or white”. Are those peptides most “cognate” for AKT, MTOR and P70 than to

any other kinase? How was that determined?

B2) Given the previous concerns, it can not be assured that these are the most involved kinases in aberrant JAK signaling. I suggest a Kinase Set Enrichment Analysis approach, with not as stringent FDR boundaries, to ensure that there is actually an enrichment in those kinases' activity in Clone 1, Clone 2 and IFN KD. Those with low FDR should be validated akin in Figure 3E. It may lead to a modification of the proposed mechanism, but, the way the data are laid out now it looks that the proteomic data are "cherry-picked" after confirming the experimental data, and not the other way around.

B3) Is figure 3D any other thing that a different representation of the highlighted peptides in 3C? If so, it must be deleted, it does not further support the previous data

B4) Finally, although not strictly related with the proteomics approach, the way data are presented it is unclear how AKT and MTOR activate JAK. If its not directly activated (phosphorylated, I recommend an in vitro kinase assay) there is a missing link between these two events. I acknowledge that this is not always easy to find but at least it must be searched, and if a direct link is not found, at least it must be mentioned specifically in the discussion.

3) Finally, the results would be more robust and convincing if they could show genomic or proteomic matching results with this preclinical research of either a cohort of melanomas primarily resistant to a-CTLA4 or melanomas with acquired resistance to a-CTLA4 due to IFN-gamma signaling loss. This would be confirmatory and would be the required piece that, from my perspective as a medical oncologist, would justify the conduction of a pilot proof-of-concept clinical trial. Although they provide TCGA data showing that IFN-gamma R1 low melanomas have bad prognosis and low expression signatures of T-cell genes, strictly speaking, with the nature of clinical annotation of TCGA this can not be blamed to JAK de-regulation and/or resistance to anti-CTLA4 agents. However, maybe the authors could further analyze their previously reported cohort, where data about CTLA4 treatment and response were available and would be a good confirmatory vehicle for the hypothesis linking JAK to treatment failure in the IFN-low melanomas.

Minor comments

-Line 43-44: Targeted therapies are also a pillar for cancer care and it should be mentioned.

-98-99 : should say "...to overcome ICB resistance in cases of IFN signaling loss"

-Line 173: "To address this, we treated scrambled and IFNgR1KO cells with supernatants harvested from cultured IFNgR1KO and did not see increase of p-JAK2" – do the authors mean "harvested from cultured Wild types or KD"? otherwise, how would they expect to see something different if they apply the autologous supernatant.

-Line 190: typo, highlighted in red

Reviewer #3 (Remarks to the Author): with expertise in melanoma, cancer immunotherapy/resistance

The overall objective of this study was the identification of novel therapeutic concepts enabling targeting of IFNGR signaling-defective melanoma cells which are resistant to immunotherapy, as previously shown by the authors.

To address this objective, the authors generated IFNGR1-ko B16 melanoma cells and demonstrated resistance of those cells to anti-CTLA-4-based immunotherapy, as expected. Seeking for strategies to eliminate IFNGR1-ko cells, the authors studied their kinome. Data analyses led to the identification of JAK1/2 as potential therapeutic targets. The authors confirmed enhanced phosphorylation of JAK1/2

in IFNGR1-ko cells in comparison to scrambled B16 cells. Based on this observation the tumor cells were treated with the JAK1/2 kinase inhibitor Ruxolitinib. The cells were responsive to the inhibitor in vitro and in vivo. Also human IFNGR1-ko A375 melanoma cells were generated and sensitivity to Ruxolitinib treatment was demonstrated. Based on their data the authors concluded that targeting JAK1/2 by Ruxolitinib could be a strategy to control IFNGR signaling-defective melanoma.

Comments to the authors:

To topic addressed by the authors is of importance and high clinical relevance. The finding of an altered kinome in IFNGR1-ko B16 cells is very interesting and of therapeutic interest. However, there are major concerns regarding the data on JAK1/2 as therapeutic target.

In vitro effective blockade of JAK1/2 kinase activity in melanoma cells can be achieved with concentrations of less than 1 μ M. The authors used 20 μ M and 30 μ M Ruxolitinib to inhibit JAK1/2 activity in B16 and A375 melanoma cells, respectively. These concentrations are far too high and most likely lead to unspecific inhibition of other kinases. The authors should

exclude unspecific effects of high inhibitor concentrations on kinases other than JAK1/2 in IFNGR1-ko and scrambled B16 cells and A375 melanoma cells.

titrate down Ruxolitinib and control phosphorylation of JAK1/2 targets after short-term and long-term Inhibitor treatment

specifically knockdown JAK1/2 in IFNGR1-ko and scrambled murine and human melanoma cells to demonstrate its critical role in cell proliferation, survival, colony forming capacity

Both, JAK1 and JAK2 show stronger phosphorylation in IFNGR1-ko compared to Scrambled B16 cells (Fig. S3A). So far, data providing evidence for enhanced JAK1/2 activity are completely lacking. Do IFNGR1-ko cells show enhanced activation of downstream STAT1-IRF1 or STAT3 transcription factors (WB for (p)STAT1, IRF1, (p)STAT3). To demonstrated downstream signaling and target gene expression the authors should carry out RNA-Seq or perform qPCR analyses for selected candidate genes in murine and human models.

IFNGR1-ko B16 cells show strongly impaired in vitro proliferative and colony formation capacity in comparison to Scrambled B16 cells (+/- Ruxolitinib). What about tumor formation in vivo in immunodeficient mice upon transplantation of identical numbers of tumors cells? In the material and method section the authors already mentioned that some experiments were carried out in RAG-/- mice, though data were not presented.

The authors claim that IFNGR1-ko cells are much more sensitive to Ruxolitinib treatment not only in vitro but also in vivo. However, comparative data on Ruxolitinib treatment of IFNGR1-ko and Scrambled B16 tumor transplants are missing. Such analyses should be carried out in immunodeficient mice, to exclude effects of inhibitor treatment on the immune cell compartment.

Data on elevated phosphorylation of JAK1/2 in human IFNGR1-ko A375 melanoma cells are completely lacking. Moreover, data presented in Fig. 6C and D do not suggest enhanced sensitivity of IFNGR1-ko cells in comparison to Scrambled A375 cells, contradictory to the data from the B16 model.

The TCGA data analysis is not very informative. The data are derived from bulk tumors and do not address tumor cell-specific IFNGR1 expression.

The data presented on enhanced cell death induction by IFNg (48h incubation, Fig. S1E) in Scrambled compared to IFNGR1-ko B16 cells are not convincing. The dot plot shows a small shift of the whole cell population under IFNG treatment but no clear Annexin positive subpopulation and no Annexin/PI positive cells. In general cell death induction by IFNg requires longer incubation times.

Reviewers' comments:

Reviewer #1 (Remarks to the Author): with expertise in cancer immunology/immunotherapy

This manuscript by Shen et al. examines IFN γ -sensing in human and mouse melanoma cells and finds a paradoxical increase in phosphorylated JAK1/2, which are the signaling kinases downstream of the IFN γ receptor. This is an intriguing finding, and suggests that either 1) other cytokines besides IFN γ are providing growth signals to the tumor cells or 2) tumor-intrinsic signaling pathways are altered to compensate for the lack of IFN γ R signaling. Given the central importance of IFN γ in regulating anti-tumor immunity in both mice and humans, this surprising finding of rewiring of tumor cell cytokine signaling is important and novel. Furthermore, drugs targeting members of the JAK family are approved for rheumatologic diseases and are being evaluated in clinical trials for several cancer indications.

We thank this reviewer for recognizing the novelty, importance, and translational potential of our study. We have now provided additional evidence supporting activation of JAK1/2 is mediated by heightened mTOR pathway (Fig. 5) but not by extrinsic signals such as type I interferon and IL-6 (Fig. 4). We have also shown that Ruxolitinib (an FDA approved inhibitor of JAK1/2) selectively suppressed IFN γ R1^{KO} but not scrambled control melanomas (Fig. 6A&B), manifesting a potential “targeted” therapy for ICB-resistant IFN γ R1^{KO} melanoma.

That said, there are major logical flaws in the manuscript as currently presented. It is not at all clear to me that the authors understand the significance of their own work, given that both the title and the abstract inappropriately focus on CD8 T cells and misrepresent the central finding of the manuscript. Furthermore, the obvious question of what signaling pathways are activated in the tumor cells was only superficially addressed, with limited number of tumor cell clones evaluated, known JAK1/2 targets like phospho-STAT1 not examined, and responsiveness to type I IFN signaling not addressed. I am still left wondering whether IFN γ R1KO cells have increased reliance on a different cytokine or growth factor, whether these tumor cells are more exquisitely sensitive to TNF α . Furthermore, the authors have not entirely convinced me that the results shown are not just artifacts from generation of clonal cell lines.

We thank this reviewer for these insightful comments. To address them, we have conducted extensive new studies and please see our point-by-point responses below for details. In brief, combining pharmacological and genetic approaches, we showed that IFN γ R1^{KO} cells were not more sensitive to type I IFN signaling (IFN- α) (Fig. 4F and Fig. S4&C) and TNF (Fig. S7D). While we could not detect p-STAT1 by immunoblotting at the baseline, on the other hand, basal p-STAT3 was also low but detectable, which showed increase in IFN γ R1^{KO} cells. These data suggest that STAT3 was a preferential target of activated JAK1/2 in IFN γ R1^{KO} cells (Fig. 3C). Our additional RNA-seq analyses (Fig. 5A&B) as well as genetic engineering studies with restoration of IFN γ R1 (Fig. 3E&F), mTOR knock-down (Fig. 5F), and IFN α R1 knock-out (Fig. 4E-G) supported that JAK1/2 activation was directly related to dysfunctional IFN- γ signaling and mediated by the augmentation of mTOR pathway but not altered extrinsic signals. Moreover, we provided evidence that Ruxo relies on T cells and host TNF signaling to exert its efficacy.

Major concerns:

1) The title is very misleading as the authors do not examine the role of JAK inhibition in T cells. Please consider rephrasing to “IFN γ -pathway deficient melanomas are sensitive to JAK inhibition” or similar title that focuses on the primary message of the paper, which is the altered JAK signaling in IFN γ R1KO

tumor cells. The one sentence summary doesn't make any sense "Tumor-intrinsic loss of IFN-g signaling confers immunotherapeutic resistance and suppresses TILs, which can be treated with by Ruxolitinib." What does "suppresses TILs" mean? Consider rephrasing to "Tumor-intrinsic loss of IFN-g signaling confers immunotherapeutic resistance which can be reversed by Ruxolitinib."

We thank this reviewer for the suggestions. What we meant by "suppress TILs" is that not only the abundance of CD8⁺ T cells in IFN γ RI^{KO} melanoma was reduced but also TILs can't be rejuvenated by ICBs. To avoid confusion, we removed this expression completely from our manuscript. In light of our new data supporting an essential role of T cells and host TNF signaling in Ruxo-induced selective suppression of IFN γ RI^{KO} melanoma (Fig. 6&7), we changed the title to "JAK inhibition selectively suppresses melanomas lacking IFN- γ -pathway". Accordingly, we have changed the one-sentence summary to "Melanomas lacking IFN- γ signaling are resistant to ICB, have constitutively active JAK1/2, and are sensitive to Ruxolitinib, whose efficacy depends on T cells and host TNF signaling."

2) The experiments looking at tumor infiltrates in IFN γ RI^{KO} tumor-bearing mice +/- anti-CTLA4 blockade are misleading. If a tumor is responding to immunotherapy and becoming smaller, then cytokine-producing CD8 effector T cells in the TILs will increase. If the tumor is not responding to anti-CTLA-4 due its known genetic loss of IFN γ RI which has already been reported to confer both primary and acquired resistance, then it will be larger and have fewer activated T cells. This is not surprising or interesting.

Generally speaking, this is a legitimate concern. However, we'd like to respectfully point out that in our previous study that showed an essential role of melanoma IFN- γ signaling in ICBs (2016, Cell), we also observed significant volumetric differences between scrambled control and IFN γ RI^{KD} melanomas after anti-CTLA-4 therapy, but there were no overt changes of cytokine production by CD8⁺ TILs and T_{reg} frequency. Further, although untreated scrambled control and IFN γ RI^{KO} melanomas were comparable in size, there was a significantly reduced frequency of CD8⁺ TILs in the latter. We think that the volumetric differences between WT and IFN γ RI^{KO} melanomas upon ICBs primarily reflected therapeutic resistance, but not the reason why infiltration and function of TILs was reduced in IFN γ RI^{KO} melanoma.

3) The experiments looking at tumor infiltrates in IFN γ RI^{KO} mice treated with Ruxolitinib are also misleading. Systemic administration of a JAK inhibitor will broadly affect cytokine signaling in all cells, including immune cells. The authors cannot conclude that Ruxolitinib treatment is acting on the increased phosphor Jak1/2 in tumor cells as its primary mechanism of action without thoroughly addressing the role of Jak1/2 in immune cells as well. Furthermore, all of the effects seen could be due to altered sensing of cytokines other than IFN γ .

We agree with this reviewer that systemic administration of Ruxo can affect all cells, including immune cells. As a matter of fact, we observed prominent modulation of TILs both in vivo and in vitro (Fig. 6). Furthermore, our new data clearly showed that Ruxo efficacy was dependent on T cells and host TNF signaling (Fig. 7); additionally, Ruxo, when used at lower reasonable concentrations (per Reviewer 2 and 3's concerns, see below), did not induce overt killing of IFN γ RI^{KO} cells (Fig. S6B&C). We therefore concluded that immunomodulation of TILs was a major functional mechanism underscoring Ruxo efficacy. Moreover, Ruxo selectively suppressed growth of IFN γ RI^{KO} melanomas (Fig. 6B) that harbored constitutively activated

JAK1/2 (Fig. 3A&B), offering a potential “targeted” therapy for IFN γ R1^{KO} melanomas. With respect to altered sensing of other cytokines, we tested two major cytokines that can engage the JAK-STAT pathway (IL-6 and type I interferons) and our results did not support an important role of these cytokines in JAK1/2 activation (Fig. 4 and S4). We also did not find greater sensitivity of IFN γ R1^{KO} cells to TNF, even when treated with very high concentrations of TNF (Fig. S7D). These data, together with no increase of p-JAK1/2 in scrambled control cells treated with IFN γ R1^{KO} supernatants containing secreted factors (Fig. 4H), suggest that JAK1/2 activation is unlikely due to altered signaling from extrinsic factors.

4) Jak1/2 are downstream of multiple cytokine receptors, including ones such as IL-6 that serve as direct tumor growth factors. The authors use conditioned media to examine whether autocrine or paracrine factors are secreted by tumor cells may account for increased Jak1/2 activity in IFN γ R1KO cells. These experiments were apparently negative, but the data shown are not convincing, and I would argue that a soluble ligand is still a likely explanation for the alterations observed. Proteomic profiling of the media and cell pellets would be more informative.

As aforementioned, IL-6 and type I interferon (IFN- α) signaling did not play an important role in JAK1/2 activation. Specifically, we showed that blocking IL-6 and IL-6R did not rescue increased p-JAK1/2; similarly, genetic deletion of IFN α R1 did not change increased p-JAK1/2, either. In further support of an essential role of aberrant mTOR signaling in JAK1/2 activation in IFN γ R1^{KO} cells, mTOR knockdown significantly suppressed p-JAK1/2 (Fig. 5F), in line with mTOR inhibition (Fig. 5E). As for proteomic analysis of the media, we'd like to point out a few limiting factors: 1. Some secreted factors are not released to the medium and instead bind to the receptors in proximity upon secretion; 2. Low levels of secreted cytokines in the medium and the limit of detection with proteomics may not detect potentially important but minute cytokines; 3. Cytokines in general are small molecules and upon sample processing, result in a small number of peptides that can be identified by mass spectrometry; 4. Unavoidable and profound interference from FBS in the culture medium would obscure the results and analyses; one may consider to use serum-free medium to improve the accuracy to identify secreted cytokines, doing that however will drastically change the growth kinetics and even the basic features of cells. For these reasons, we respectfully reason that proteomics of the medium may not provide the answer to pinpointing the importance of paracrine vs cell-intrinsic signaling alterations. That said, with our newly added data, we are comfortable to state that augmented mTOR pathway in IFN γ R1^{KO} cells is a major cell-intrinsic mechanism mediating JAK1/2 activation.

5) The phosphosite profiling is very interesting and should be explained more clearly to the non-specialist audience. It is unclear to me why STAT proteins, known targets of JAK1 and JAK2, were not identified by this analysis. Could the authors show pSTAT1 by immunoblotting? As this is the key transcription factor activated by IFN γ R, STAT1 is of particular importance whether it is changed or unchanged.

Following this suggestion, we have provided a detailed explanation of phosphosite profiling in this revision (please refer to the text). Regarding the no-show of STAT proteins in our proteomics studies, a main reason is that our sample preparation for proteomics using TiO₂ beads enriched peptides with serine and/or threonine phosphorylation, whereas STAT proteins are activated primarily by tyrosine phosphorylation. Another factor to consider is that the enrichment of

phosphorylated peptides and mass spectrometry analysis depend on peptide abundance and protein size. As aforementioned, basal levels of p-STAT1 and p-STAT3 are very low in melanoma cells. Altogether, these facts explain why they were not detected by our proteomic analysis. Although we detected increased p-STAT3, we were unable to detect p-STAT1 by immunoblotting, due to its extremely low level at the baseline. While it remains to be determined if p-STAT1 is increased or not in IFN γ R1^{KO} cells by using more sensitive method other than Western blot, increased p-STAT3 nevertheless indicated activation of the JAK1/2-STAT3 axis.

6) The data from TNFa blockade treated mice are intriguing, and suggest a clear mechanism of IFN γ R1KO tumor cells having increased sensitivity to TNFa-mediated death pathways. Please examine in vitro the relative TNFa sensitivity of control and KO cell lines.

Following this suggestion, we treated control and IFN γ R1^{KO} cells with various doses of TNF; we did not see much killing even from very high doses of TNF (i.e., 10000 U/mL) (Fig. S7D). Given these results and the abolition of Ruxo efficacy in TNF^{-/-} mice (Fig. 7B&C), mice treated with blocking antibodies against TNF (Fig. S7C), and mice depleted of either CD4⁺ and CD8⁺ T cells (Fig. 7A), we reason that Ruxo therapy of IFN γ R1^{KO} melanomas engage immune-related mechanisms (T cells and TNF signaling), which in turn control tumor growth (Fig. 6&7).

7) Figure 5I: Please include Ruxolitinib, TNFa, and Ruxolitinib+TNFa treatment of control B16 tumors as well for a comparison to the IFN γ R1KO tumor growth curves.

We have included data from mice bearing scrambled control tumors treated with Ruxo, which showed that Ruxo did not work in scrambled control melanomas (Fig. 6A). We therefore did not pursue Ruxo treatment further in control B16 tumors. Further, comparison of growth kinetics of scrambled control and IFN γ R1^{KO} melanoma did not reveal overt differences in Rag-1^{-/-} mice (Fig. 1G) and B6 mice (Fig. 1H).

8) Please justify the use of clones rather than populations of IFN γ R1KO cells. Single cell expanded clones can have altered growth behavior that deviates from the parental cell line merely by chance, and not necessarily related to the specific gene deleted. If clones were used, a minimum of 4 clones must be evaluated, or a single knockout clone of interest must be reconstituted with IFN γ R1 to show that phospho JAK1/2 levels are restored to baseline.

As known, CRISPR-Cas9 technology relies on the binding of CAS9 and sgRNAs to targeted DNA sites, which then introduce double strand breaks (DSB) and cause deletion of genes of interest. While the majority of successfully transduced cells will retain the deletion status, a rare number of cells can completely repair DSB to restore the genes or only have genes shortened by a few base pairs that still retain partial functionality. To rule these possibilities out, we decided to use single clones as “clean” systems to unequivocally dissect the role of tumor-intrinsic IFN- γ signaling in ICB response and TIL modulation. Following this reviewer’s suggestion, we re-expressed IFN γ R1 in IFN γ R1^{KO} cells (Fig. 3E), which largely rescued increased p-JAK1/2 (Fig. 3F), establishing a direct link of JAK1/2 activation to lack of IFN- γ signaling.

Minor concerns:

1) Line 113 “scrambled cells” is an odd phrase. Consider “scramble control cells” instead.

We have changed scrambled cells to “scrambled control cells” throughout.

2) Lines 347-351: “We reason that if downregulation of MHC II on IFNgR1KO melanoma plays a dominant role in shaping TILs, immunosuppression of TILs would ensue; conversely, if reduced expression of PD-L1 on tumor cells is important, immunostimulation of TILs would occur. The overall reduced abundance and function of TILs support downregulation of MHC but not PD-L1 is the more dominant factor in shaping TILs in the absence of tumor-intrinsic IFN-g signaling”. This is an overly simplistic explanation. Please do not refer to suppression or stimulation of TILs as a bulk entity as TILs encompass a variety of distinct cell types with distinct functions.

We have rephrased, as follows,

“However, it is unknown whether tumor-intrinsic IFN- γ signaling modulates TILs. On one hand, IFN- γ , by upregulating MHC molecules and activating tumor antigen processing and presentation machinery^{60, 61, 62, 63, 64}, promotes anti-tumor immunity; on the other hand, it can also suppress anti-tumor immunity by inducing various regulatory mechanisms such as PD-L1 upregulation in stromal and tumor cells⁶⁵. We observed markedly reduced expression of both MHC molecules and PD-L1 in IFN γ R1^{KO} melanoma, although the former was more pronounced. Our study corroborated an early pioneering study by Bob Schreiber and colleagues, who demonstrated that IFN γ R1 truncation in methA fibrosarcoma decreased tumor immunogenicity and responsiveness to LPS therapy⁵⁹. Although our results suggest that lack of inducible PD-L1 upregulation in IFN γ R1^{KO} melanomas has a seemingly nonessential role in shaping TILs, this is likely a context-dependent finding, as incongruous results have been reported for the importance of tumor PD-L1 upregulation in anti-tumor immunity^{66, 67, 68}.”

3) Figure 1A: Why are the bands pictured not at the same height across the 4 samples? Were these run on different gels Supplemental Figure 1C should be added to main Figure 1 for comparison.

We have fixed it. These were run on the same gels, but these samples were not loaded to adjacent wells. For the clarity of data presentation, we manually placed them next to each other. We have moved the original Supplemental Fig. 1C together with the original Fig. 1A (now, Fig. 1B).

4) Figure 1E: Please report actual tumor measurements in cubic mm rather than fold change. If the IFNgR1KO clone tumors grow differently from the parental line, this is important information to convey.

We have shown Tumor sizes in mm³ in new Fig. 1G&H.

5) Figure 2 flow plots are shown as dots plots, a heatmap, and contour plots. Please pick one graphical display format to use consistently throughout.

Thanks for this suggestion. If it is OK with this reviewer, we would like to keep the original formats, as they better presented our key point. For example, Fig. 2C showed both frequency and color-coded GMFIs.

6) Figure 2: Tumor size at time of harvest needs to be shown. CD8 T cells vary based on tumor size, so if the IFNgR1KO tumors, which are unresponsive to anti-CTLA4 therapy, were larger at the time of harvest, then the results presented here are entirely expected and could be moved to supplemental.

Please refer to our response to Major concerns #2 for detailed explanation that volumetric differences were not the cause of our findings in TILs. We'd like to point out that altered T cell infiltration, cytokine production, and T_{reg} abundance in $IFN\gamma R1^{KO}$ tumors are very novel, distinct from our previous study (2016 Cell) using the $IFN\gamma R1^{KD}$ model. To the best of our knowledge, this is the first study to show an important role of tumor-intrinsic $IFN-\gamma$ signaling in shaping TILs. For this reason, we'd like to keep Figure 2 as an important figure.

7) Figure 4 and Supplemental Figure 6: It appears from the data shown that all melanoma cells are sensitive to Ruxolitinib, not just the $IFN\gamma R1^{KO}$ cells. Please change the figure titles and accompanying text to not mislead the readers into thinking that $IFN\gamma R1^{KO}$ cells are preferentially sensitive.

We removed those figures, because of serious concerns from Reviewers 2 and 3. Specifically, they were concerned with the super-high concentrations of Ruxolitinib (20 and 30 μM) used in those studies, which would well induce non-specific killing of tumor cells. Reviewer 2 further stressed that these high doses of Ruxo can't be achieved in vivo (rather, 1000-fold less). Following their suggestions, we tested Ruxolitinib at much lower doses (1, 10, 100, and 1000 nM) and did not see no overt killing, despite potent suppression of JAK-STAT.

Reviewer #2 (Remarks to the Author): with expertise in cancer biology, phosphoproteomics

NCOMMS-21-25916

Hongxing Shen and Lewis Shi present a manuscript entitled "JAK inhibition reprograms intratumoral T cells and bypasses tumor intrinsic immunotherapeutic resistance"

Dr. Shi pioneered the discovery of IFN gamma signaling as a mediator to immune checkpoint inhibitors. In this manuscript, as a senior author, his group reports mechanistic cues on this phenomenon, mainly consisting on an aberrant signaling axis involving JAK, which in turns regulates TILs infiltration rate and their functionality. This axis is therapeutically targetable, and thus the information contained in this manuscript could provide a basis for a proof of concept clinical trial. The described mechanisms is new and sheds light over a significant problem in oncology, and thus, it could warrant publication in Nature Communications. However, a few major observations impede the publication of the manuscript as it present form, particularly regarding the description of the mechanisms and the proteomics approaches.

We thank this reviewer for this high evaluation of our study.

1) Mechanistic explanation. The main point here is that JAK1/2 de-regulates TNF signaling and this affects negatively to the infiltration and activation of TILs. This is well shown and I accept this part. However, at the same time, the authors defend that Ruxolitinib has cell-autonomous effects (whole Fig 4 data). This, on itself, is counter-intuitive and unnecessary (?). If the effect is so strong in a cell-autonomous manner, then, what is the role of JAK in vivo? Kill the cells? Or reactivate the tumor infiltrating lymphocytes? It can not be claimed the latter if the tumor cell specific is so strong. Experiments depleting CD4 and/or CD8 cells (or their mediators IFN , TNF, IL2,... shown in Fig 5) in vivo should allow weighting whether the effect is more Ruxolitinib-mediated cell killing or more Ruxolitinib-mediated TIL infiltration and invigoration. If it is because of cell killing, the claim of infiltration and invigoration invalidates the whole point of the manuscript. However, an alteranative hypothesis is possible, which is that the authors used in vitro 20 micromolar of Ruxolitinib, which is something that

can not be achieved in vivo in the stroma (1000-fold less, actually) and it may be killing cells independently of JAK (off target) because of ultra high dose. If this is the case (test killing non-JAK-dependent models, or JAK Kos, or test lower dose), then, the in vitro data add nothing and should be removed. If the claim is to defend that this is the reason ICB fails, authors do not need to show any cell-autonomous effect of the drug that reverses the ICB failure, they only need to show that they rescue ICB activity, and actually this would be better supported in absence of direct activity of the drug in the tumor cells (cleaner effect) (. I suggest clarifying this point in vivo, and formally excluding cancer cell killing by Ruxolitinib (non-specific, or at least, not ICB failure-rescue mediated). Same for the in vitro data with the human cell line.

We thank this reviewer for the constructive comments. We agree with this reviewer that Ruxo concentrations used in our previous in vitro experiments were too high (20 or 30 μ M), which can't be achieved in vivo in the tumor stroma; as this reviewer pointed out, it is probably "1000-fold less". We have therefore removed all the old in vitro data with mouse and human cell lines. In addition, we have tested lower concentrations that are physiologically relevant. Our new data showed that at concentrations (10-1000 nM) that potently suppressed JAK-STAT (Fig. S6A), no overt cell killing (Fig. S6B), and suppression of colony-forming units (Fig. S6C) were seen. Moreover, following this reviewer's suggestion, we depleted CD4⁺ or CD8⁺ T cells with neutralizing antibodies and employed TNF^{-/-} mice to test the role of Ruxo-mediated modulation of TILs and functional invigoration (TNF production) in this process. Clearly, both T cell depletion (Fig. 7A) and lack of host TNF signaling (Fig. 7B&C) abrogated therapeutic effects of Ruxo (Fig. 6B). These new data greatly strengthen our study and establish Ruxo-driven immunomodulation of TILs as a major underlying mechanism for Ruxo efficacy.

2) Proteomics data generation – Involvement of JAK1/2 and serine/threonine kinases in tumors with low IFN gamma R1.

a) Data for detecting tyrosine kinases aberrantly activated

a.1) The chip approach is not a widespread technique and it should be more thoroughly described. Both technically and analytically. If there are text length limitations, it should go in the material and methods. Primary, and more importantly, secondary analysis, should be thoroughly described (i.e., "how they come up with these kinases"). Chips have many issues for proteomics (sensitivity, specificity, semiquantitative at best), although less when using a limited number of targeted proteins, and should be discussed. I would be more convinced from a mass-spectrometry guided approach following a tyrosine phosphorylation enrichment step.

We have provided the relevant details about the chip-based kinomic analysis in the method of "kinomic analysis". Due to the length, we did not list it here (please refer to main text).

a.2). Since many (26) kinases were observed as enriched (and the number of tyrosine kinases is not that large) I suggest at least discarding the involvement of those with high scores, other than JAK, in supplementary figures similar to 3B, blotting other kinases. It may occur, due to the network nature of these signaling events, that there are more upstream players that, when JAK is therapeutically targeted, keep feeding the network, lacking a final positive outcome.

Following this suggestion, we have examined the three kinases with high mean final scores (an indication of specificity) (Fig. S3A). While basal p-EphA3 could be detected by immunoblotting, basal levels of p-ZAP70 and p-Syk were extremely low (Fig. S3C&D). Overall, they did not show

significant increases in $IFN\gamma R1^{KO}$ cells. Given these results and our finding of activated JAK1/2 being the central node in the active PTK network (Fig. 3A), we focused on JAK1/2. While it is a legitimate concern that other upstream players may feed into this network and possibly attenuate therapeutic effects from JAK1/2i, we did show that Ruxo significantly suppressed $IFN\gamma R1^{KO}$ melanomas in vivo. While other upstream players await to be identified, we found mTOR is a major upstream regulator of JAK1/2. We therefore treated $IFN\gamma R1^{KO}$ cells with both Ruxo (1 μM) and Rapa (an mTOR inhibitor, 50 nM) and did not find greater killing, as compared to Rapa alone. Note: as described above, 1 μM of Ruxo did not induce cell killing, in spite of potent inhibition of JAK1/2. Given these were negative results, we did not include them in this revision. But, if this reviewer prefers, we can provide them.

a.3.) The analytic approach overlying genomics and proteomics to pinpoint JAK (3a) has to be described in more detail, since most readers will not be familiar with it.

A more detailed description about the kinomic study including analysis has been included in the methods for "Kinomic analysis".

b) Data for detection of aberrantly active serine/threonine kinases

b1) Although the mass spectrometry approach looks state of the art, only the explanation for the primary analysis (i.e., peptide identification, quantitation, etc) is provided. I particularly miss information about the secondary analysis. For example:

-How were the proteins shown in Figure 3C selected? How come there are not hundreds, if not thousands, of other proteins regulated in clone 1, clone 2 or KD versus scrambled? The cut-off score for showing only those should be shown, explained and justified.

We have provided further details on the process of how the proteins and peptides from our proteomic studies were selected. Briefly, we chose phosphorylation sites based on the following criteria: 1) P values were less than 0.05 for $IFN\gamma R1^{KO}$ vs scrambled control cells. 2) Kinases corresponding to these phosphosites had been experimentally identified. 3) Phosphorylation was increased in $IFN\gamma R1^{KO}$ cells. There were 19 increased phosphorylation sites meeting these criteria. The kinases mediating these 19 phosphosites were used for pathway enrichment analysis. The Uniprot IDs (mouse) for these kinases were mapped to KEGG IDs (mouse), which were then used for pathway enrichment at KEGG website.

-The previous question entails with the fact that many other proteins not shown there may map to other kinases implicated in the loop. Why were they filtered out?

We provided all the phosphosites with our prior submission. Of note, our kinomic analyses focused on kinases whose activities are primarily regulated by tyrosine phosphorylation, given their essential roles in the JAK-STAT pathway, whereas the proteomic analyses were carried out to identify the serine/threonine kinases that have been known to intricately interact with PTKs, as an effort to identify regulators of the JAK-STAT axis. Thus, the overlapped kinases between these two systems were limited, although the proteomic analysis can pick up some serine/threonine kinases that were also phosphorylated at their tyrosine residues. With respect to how the phosphosites were chosen, please refer to our response above as well as our response to Major Concerns #5 from Reviewer 1.

-The highlighted phosphopeptides in red are substrates of AKT, MTOR and p70S6K according to the authors. How was this concluded? Phospho-peptides actually have a long consensus sequence that can in fact be target of many kinases, and have higher affinity or lower affinity to a diverse number of kinases, it is not “black or white”. Are those peptides most “cognate” for AKT, MTOR and P70 than to any other kinase? How was that determined?

This is an excellent point. We completely agree with this reviewer that phospho-peptides can actually be mediated by different kinases. It is rare to have “black or white” cognate phosphopeptides for certain kinases. Upon further research into this, we de-colored those peptides (Fig. 5C); instead, we reason that if activation of kinases in a pathway can explain most of the identified phosphosites (as in our case, 11 out of 19 and 10 out of 19 are connected to the PI3K-AKT and the mTOR pathways, respectively) (Fig. S5A), a great level of confidence can be reached to conclude that these pathways are activated in IFN γ R1^{KO} cells. Furthermore, our RNA-seq analysis also identified PI3K-AKT and mTOR in IFN γ R1^{KO} cells (Fig. 5A&B), which were confirmed directly by WB (Fig. 5D). Moreover, knockdown of mTOR in IFN γ R1^{KO} cells greatly rescued JAK1/2 activation (Fig. 5F). Therefore, these result together convincingly established the IFN γ R1^{KO} \rightarrow mTOR \rightarrow JAK1/2 axis in melanoma (Fig. 3&5).

B2) Given the previous concerns, it can not be assured that these are the most involved kinases in aberrant JAK signaling. A suggest a Kinase Set Enrichment Analysis approach, with not astringent FDR boundaries, to ensure that there is actually an enrichment in those kinases' activity in Clone 1, Clone 2 and IFN KD. Those with low FDR should be validated akin in Figure 3E. It may lead to a modification of the proposed mechanism, but, they way the data are laid out now it looks that the proteomic data are “cherry-picked” after confirming the experimental data, and not the other way around.

Please refer to our above response to B1 for the rationales and detailed explanation of how those phospho-peptides were chosen and how the pathway enrichment analysis was performed. Our focus on the PI3K-AKT-mTOR pathway was formed based on our overall data. Importantly, we showed that augmentation of mTOR is a major regulator of JAK1/2 activation in IFN γ R1^{KO} cells using biochemical (direct immunoblotting for p-JAK1/2 and p-S6/AKT/4E-BP-1) (Fig. 3B and Fig. 5D), pharmacological (mTOR inhibition by Rapa) (Fig. 5E) and genetic (mTOR^{KD}) approaches (Fig. 5F). That said, it is possible that other signaling pathways can be involved in JAK1/2 activation, pending further examinations. We deposited all the phosphor-peptides with significant changes in scrambled vs IFN γ R1^{KO} cells in our original submission, which can be mined by readers for their potential broad interests.

B3) Is figure 3D any other thing that a different representation of the highlighted peptides in 3C? If so, it must be deleted, it does not further support the previous data.

We have removed the original Fig. 3D.

B4) Finally, although not strictly related with the proteomics approach, the way data are presented it is unclear how AKT and MTOR activate JAK. If its not directly activated (phosphorylated, I recommend an in vitro kinase assay) there is a missing link between these two events. I acknowledge that this is not always easy to find but at least it must be searched, and if a direct link is not found, at least it must be

mentioned specifically in the discussion.

We really appreciate the suggestion. Addressing this, we have cited a recent study reported a direct mutually interactive loop between the mTOR and JAK pathways in colorectal cancer cells³³. To assess how they interact and regulate each other in melanoma, we first took a pharmacological approach by treating cells with Rapamycin (Rapa), a well-established mTOR inhibitor and found that Rapa profoundly suppressed p-JAK2 (Fig. 5E); conversely, inhibition of JAK1/2 with Ruxo did not change p-4E-BP1 in IFN γ R1^{KO} cells (Fig. S5B), placing mTOR upstream of JAK1/2. To specifically assess the role of mTOR in JAK1/2 activation in IFN γ R1^{KO} cells, we knocked down mTOR using shRNAs (mTOR^{KD}) (Fig. 5F). Similar to Rapa treatment, mTOR^{KD} also significantly reduced p-JAK1/2 and more importantly rescued JAK1/2 activation at least partially in IFN γ R1^{KO} cells (Fig. 5F). Together, these results establish heightened mTOR as a major upstream regulator of JAK1/2 activation in IFN γ R1^{KO} melanoma.

Following the Reviewer's suggestion, we included the following in Discussion.

"As we recently described²³, as principal gatekeepers of various cellular signaling pathways, JAK1/2 are delicately regulated at different levels including post-translational modifications, inhibitory function of the pseudokinase domain, as well as many regulators including phosphatases, Protein Inhibitors of Activated STAT (PIAS) that inhibit STAT-DNA binding, and suppressor of cytokine signaling (SOCS, a negative feedback mechanism)⁸². It would be interesting to investigate how IFN γ R1^{KO}-mTOR axis affects these regulatory mechanisms in the future, especially the activity of protein tyrosine phosphatases (PTPs) to mediate activation of JAK1/2."

3) Finally, the results would be more robust and convincing if they could show genomic or proteomic matching results with this preclinical research of either a cohort of melanomas primarily resistant to a-CTLA4 or melanomas with acquired resistance to a-CTLA4 due to IFN-gamma signaling loss. This would be confirmatory and would be the required piece that, from my perspective as a medical oncologist, would justify the conduction of a pilot proof-of-concept clinical trial. Although they provide TCGA data showing that IFN-gamma R1 low melanomas have bad prognosis and low expression signatures of T-cell genes, strictly speaking, with the nature of clinical annotation of TCGA this can not be blamed to JAK de-regulation and/or resistance to anti-CTLA4 agents. However, maybe the authors could further analyze their previously reported cohort, where data about CTLA4 treatment and response were available and would be a good confirmatory vehicle for the hypothesis linking JAK to treatment failure in the IFN-low melanomas.

The database reported in our previous paper (2016 Cell) was derived from whole exome sequencing and did not contain gene expression data, preventing us from performing the suggested analysis by the reviewer. That said, to circumvent this, we analyzed both the TCGA database of IFN γ R1^{Low} (thus attenuated IFN- γ signaling) vs IFN γ R1^{High} skin cutaneous melanomas (SKCMs), as well as the TCGA database of uveal melanomas (UVMs) vs SKCMs. Interestingly, UVMs are known to be more resistant to ICBs than SKCMs. Detailed analyses of UVMs vs SKCMs revealed reduced expression of IFN γ R1 as well as T cell signature genes (Fig. 2F), in line with our previous results showing downregulation of T cell signature genes in

IFN γ RI^{Low} SKCMs vs IFN γ RI^{High} SKCMs (Fig. 2E). These results support that IFN- γ signaling in human melanoma can shape TILs.

To gain an idea of mTOR and JAK1/2 activation in human melanoma, we took an alternative approach, because phosphorylation data on JAK1/2 and mTOR were not available in these public databases, precluding a direct examination of their functional activation. Based on studies in other tumor types (we were unable to find related studies in melanoma), we constructed a list of target genes of mTOR and JAK1/2 (i.e., ENO1, FASN, FKBP4, ODC1, GADD45A, JUNB, and VEGFA). To specifically analyze their expression in melanoma cells, we deconvoluted the TCGA database of SKCMs and the GSE78220 database of melanoma patients treated with anti-PD-1 that were derived from bulk tumor samples, using a panel of melanoma cell-specific genes⁴⁵. While not all the genes showed significant changes in melanoma cells, we did observe upregulation of ENO1, FASN, and FKBP4, as well as downregulation of GADD45A in IFN γ RI^{Low} SKCMs (Fig. S5C); on the other hand, patient melanomas that were resistant to anti-PD-1 exhibited significant upregulation of ENO1, FKBP4, ODC1, and VEGFA (Fig. S5D). The other genes exhibited expected increases/decrease but did not reach statistical significance (Fig. S5C&D). In spite of the differences of affected genes between the TCGA and GSE78220 databases, two genes (ENO1 and FKBP4) were consistently upregulated in both IFN γ RI^{Low} SKCMs and anti-PD-1 non-responders, suggesting some degree of activation of mTOR and JAK1/2.

-Line 43-44: Targeted therapies are also a pillar for cancer care and it should be mentioned.

We originally grouped targeted therapy into chemotherapy. With this comment, we have singled targeted therapies out as a separate pillar.

-98-99 : should say “...to overcome ICB resistance in cases of IFN signaling loss”

We have changed this.

-Line 173: “To address this, we treated scrambled and IFN γ RI^{KO} cells with supernatants harvested from cultured IFN γ RI^{KO} and did not see increase of p-JAK2” – do the authors mean “harvested from cultured Wild types or KD”? otherwise, how would they expect to see something different if they apply the autologous supernatant.

Sorry for the confusion. We have rephrased this to “Lastly, to interrogate the potential regulation of JAK1/2 activation by other factors secreted by IFN γ RI^{KO} cells into the supernatant (SN) (other cytokines, growth factors, extracellular vesicles, etc.), we treated scrambled control cells with SNs harvested from IFN γ RI^{KO} cultures for 24h. This did not induce increase of p-JAK2 (Fig. 4H), suggesting that secreted factors may not be important. Further, increased p-JAK2 in IFN γ RI^{KO} cells persisted, regardless of the SNs from IFN γ RI^{KO} and scrambled control cultures used, suggesting this is more of a cell-intrinsic event.”

-Line 190: typo, highlighted in red

We have changed this.

Reviewer #3 (Remarks to the Author): with expertise in melanoma, cancer immunotherapy/resistance

The overall objective of this study was the identification of novel therapeutic concepts enabling targeting of IFNGR1 signaling-defective melanoma cells which are resistant to immunotherapy, as previously shown by the authors.

To address this objective, the authors generated IFNGR1-ko B16 melanoma cells and demonstrated resistance of those cells to anti-CTLA-4-based immunotherapy, as expected. Seeking for strategies to eliminate IFNGR1-ko cells, the authors studied their kinome. Data analyses led to the identification of JAK1/2 as potential therapeutic targets. The authors confirmed enhanced phosphorylation of JAK1/2 in IFNGR1-ko cells in comparison to scrambled B16 cells. Based on this observation the tumor cells were treated with the JAK1/2 kinase inhibitor Ruxolitinib. The cells were responsive to the inhibitor in vitro and in vivo. Also human IFNGR1-ko A375 melanoma cells were generated and sensitivity to Ruxolitinib treatment was demonstrated. Based on their data the authors concluded that targeting JAK1/2 by Ruxolitinib could be a strategy to control IFNGR1 signaling-defective melanoma.

Comments to the authors:

The topic addressed by the authors is of importance and high clinical relevance. The finding of an altered kinome in IFNGR1-ko B16 cells is very interesting and of therapeutic interest. However, there are major concerns regarding the data on JAK1/2 as therapeutic target.

We really appreciate this reviewer for the positive comments on the importance and high clinical relevance of our study.

In vitro effective blockade of JAK1/2 kinase activity in melanoma cells can be achieved with concentrations of less than 1 μM . The authors used 20 μM and 30 μM Ruxolitinib to inhibit JAK1/2 activity in B16 and A375 melanoma cells, respectively. These concentrations are far too high and most likely lead to unspecific inhibition of other kinases. The authors should exclude unspecific effects of high inhibitor concentrations on kinases other than JAK1/2 in IFNGR1-ko and scrambled B16 cells and A375 melanoma cells.

This is an excellent point. Addressing this comment and comments from Reviewer 2, we have conducted additional experiments with lower concentrations of Ruxo. In keeping with this reviewer's assessment, 1 μM of Ruxo completely inhibited JAK-STAT pathway and potent inhibition of JAK-STAT was already observed at 10 nM (Fig. S6A). Unfortunately, we did not observe overt killing of $\text{IFN}\gamma\text{R1}^{\text{KO}}$ cells by Ruxo at these physiologically relevant concentrations (Fig. S6B&C), suggesting Ruxo suppression of $\text{IFN}\gamma\text{R1}^{\text{KO}}$ melanoma might not be due to direct cell killing. Given these results and this Reviewer's concern of non-specific effects from the "far too high" concentrations of Ruxo (20 and 30 μM), we have completely removed those in vitro data.

titrate down Ruxolitinib and control phosphorylation of JAK1/2 targets after short-term and long-term Inhibitor treatment.

Please refer to our above responses.

specifically knockdown JAK1/2 in IFNGR1-ko and scrambled murine and human melanoma cells to demonstrate its critical role in cell proliferation, survival, colony forming capacity.

Since we didn't see much change to the total amount of JAK1/2 (Fig. 3B), it was more of the activity enhancement of JAK1/2 in IFN γ RI^{KO} cells, which was mediated by augmented mTOR signaling (Fig. 5). We therefore knocked down mTOR, which significantly reduced p-JAK1/2 in IFN γ RI^{KO} cells and at least partially restored increased p-JAK1/2 (Fig. 5F), consistent with our Rapamycin data. As discussed above, we have removed all the old in vitro cell growth data, including human cell data. As laid out in our response to #3 comment from Reviewer 2, we have conducted additional in-depth bioinformatics studies, which supported clinical relevance of our preclinical findings (Fig. 2E&F, Fig. S2F, and Fig. S5C&D).

Both, JAK1 and JAK2 show stronger phosphorylation in IFNGR1-ko compared to Scrambled B16 cells (Fig. S3A). So far, data providing evidence for enhanced JAK1/2 activity are completely lacking. Do IFNGR1-ko cells show enhanced activation of downstream STAT1-IRF1 or STAT3 transcription factors (WB for (p)STAT1, IRF1, (p)STAT3). To demonstrated downstream signaling and target gene expression the authors should carry out RNA-Seq or perform qPCR analyses for selected candidate genes in murine and human models.

We have included new data showing increased p-STAT3 in IFN γ RI^{KO} cells (Fig. 3C), but we could not detect p-STAT1. Nevertheless, these data at least support activated JAK1/2-STAT3 pathway. Following this Reviewer's suggestion, we conducted RNA-Seq studies, which revealed the PI3K-AKT and mTOR were among the major pathways activated in IFN γ RI^{KO} cells (Fig. 5B). Unfortunately, unbiased analysis of RNA-seq data using all the differentially expressed genes (DEGs) did not show JAK-STAT pathway was a top hit, perhaps because of too few target genes in this pathway were impacted, as a result of very low basal levels of p-STAT1 and p-STAT3 in IFN γ RI^{KO} cells, as discussed above.

IFNGR1-ko B16 cells show strongly impaired in vitro proliferative and colony formation capacity in comparison to Scrambled B16 cells (+/- Ruxolitinib). What about tumor formation in vivo in immunodeficient mice upon transplantation of identical numbers of tumors cells? In the material and method section the authors already mentioned that some experiments were carried out in RAG-/- mice, though data were not presented.

In response to this Reviewer's and Reviewer 2's concerns over the high doses of Ruxo used in our prior submission, we have removed all the old in vitro data. Importantly, we did not observe growth defect of IFN γ RI^{KO} melanoma in Rag-1^{-/-} as well as in B6 mice (Fig. 1G&H), indicating a largely comparable tumor formation capacity to scrambled control melanoma.

The authors claim that IFNGR1-ko cells are much more sensitive to Ruxolitinib treatment not only in vitro but also in vivo. However, comparative data on Ruxolitinib treatment of IFNGR1-ko and Scrambled

B16 tumor transplants are missing. Such analyses should be carried out in immunodeficient mice, to exclude effects of inhibitor treatment on the immune cell compartment.

As pointed out by this reviewer and Reviewer 2, Ruxo concentrations previously used were too high to be achievable in tumor stroma (actually, ~1000 fold less). We have therefore removed all the original in vitro data. For this revision, using physiologically relevant concentrations of Ruxo in this revision, we did not observe overt killing of either scrambled control or IFN γ RI^{KO} cells (Fig. S6B&C), let alone differential sensitivity to Ruxo. In addition, we have provided in vivo data showing that Ruxo selectively suppressed IFN γ RI^{KO} but not scrambled control melanoma, in a T cell- and host TNF signaling-dependent manner (Fig. 7). This points to an essential role of immune-mediated mechanisms underlying Ruxo efficacy (Fig. 6&7).

Data on elevated phosphorylation of JAK1/2 in human IFNGR1-ko A375 melanoma cells are completely lacking. Moreover, data presented in Fig. 6C and D do not suggest enhanced sensitivity of IFNGR1-ko cells in comparison to Scrambled A375 cells, contradictory to the data from the B16 model.

As discussed above, to address constructive comments from all three reviewers, we have conducted experiments using lower and physiologically relevant concentrations of Ruxo that potently suppressed JAK-STAT pathway; these new data did not show any overt killing of either scrambled control or IFN γ RI^{KO} cells (Fig. S7B&C), we have therefore removed the in vitro data, including human melanoma data.

The TCGA data analysis is not very informative. The data are derived from bulk tumors and do not address tumor cell-specific IFNGR1 expression.

This is a great point. We have performed deconvolution of the TCGA database using a panel of well-defined melanoma signature genes, which provides a more accurate assessment of tumor cell-specific gene expression (IFNGR1 in Fig. 2E&F, and Fig. S5C&D). However, for T cell signature genes, we need to use the gene expression data from bulk tumors (other genes in Fig. 2E&F).

The data presented on enhanced cell death induction by IFN γ (48h incubation, Fig. S1E) in Scrambled compared to IFNGR1-ko B16 cells are not convincing. The dot plot shows a small shift of the whole cell population under IFN γ treatment but no clear Annexin positive subpopulation and no Annexin/PI positive cells. In general cell death induction by IFN γ requires longer incubation times.

We have provided new data after treating cells with IFN- γ for 96 h, which showed distinct populations of Annexin V⁺ and Annexin-V⁺7-AAD⁺ cells (Fig. S1C).

REVIEWER COMMENTS

Reviewer #1 (Remarks to the Author):

In this revised version, the authors clarify several important points regarding the mechanism of constitutive JAK1/2 phosphorylation in IFNGR1^{-/-} B16 cells. However, they then also show that the effects of Ruxo are not on the cancer cells themselves, but rather on the host immune cells. This represents a reversal of their original conclusion, which is fine – that's what good peer review should occasionally do. Perhaps counterintuitively, it seems that blockade of JAK1/2 – critical mediators of IFNg signaling – in T cells may actually reduce Tregs and increase IFNg, TNF, and IL-2 production. Apparently (I am extrapolating here somewhat) IFNg signaling in tumor cells is a critical part of the efficacy of checkpoint blockade, but when tumor-intrinsic IFNg signaling is rendered obsolete, then the negative effects of IFNg signaling on T cells outweigh the positive benefits of IFNg. This is somewhat interesting, but not fully explored in the data shown here. The issue of JAK1/2 signaling in tumor cells remains unresolved. If the increased JAK1/2 is not important for the therapeutic effect of Ruxo, what is it doing? Was this entire exploration of JAK1/2 in tumor cells all for naught? Perhaps the IFNGR1^{-/-} tumor cells are secreting chemokines or something that is responsible for the baseline alternations in T cell infiltrates. This could be more clearly explained in the Discussion.

The authors use ref 71 to claim an OS benefit in pancreatic cancer upon treatment with capecitabine and ruxo. This approach unfortunately failed for futility in two Phase III trials (PMID: 29508247). Please delete this sentence and reference.

Reviewer #2 (Remarks to the Author):

The authors have addressed all my concerns; I think that the manuscript deserves publication now. Congratulations for the great work .

Reviewer #3 (Remarks to the Author):

Shen et al. aimed to therapeutically target IFNg signaling-defective melanomas. To do so they generated IFNGR1 ko B16 melanoma cells. B16-IFNGR1ko tumors were non-responsive to anti-CTLA-4 therapy, in contrast to B16 WT tumor. B16-IFNGR1ko tumors showed low-level T cell infiltrates and poor T cell activation. Treatment with Ruxolitinib impaired the growth of IFNGR1ko B16 tumors, and this therapeutic effect was dependent on TNFa.

Comment of the reviewer:

In the first part of their study the authors demonstrated that B16 IFNGR1ko cells do not respond to IFNg and that IFNGR1ko tumors are no longer sensitive to anti-CTLA-4 therapy. This is expected based on previous studies from the authors and others. In the following they characterized the B16 IFNGR1ko tumor cells, the T cell infiltrate of B16 IFNGR1ko tumors and developed a therapeutic approach against IFNg-resistant tumors.

It is of strong clinical interest to define strategies to overcome therapy resistance of IFNG-signaling defective tumors. However, with respect to the study by Shen et al, there are several concerns regarding data interpretation and conclusions.

B16 tumor are characterized by low MHC class I expression, as shown in multiple studies including a recent work by Ribas and colleagues (PMID: 33055240). The efficacy of different T cell-based therapies is dependent on the upregulation of MHC class I expression in B16 tumors which does not take place in IFNG-resistant tumors. Moreover, it is well established that MHC class I-negative/low tumors show only low or no T cell infiltrates. Thus, low T cell numbers and low T cell activation in B16 IFNGR1ko tumors could just be a consequence of low MHC class I expression on tumor cells. To address this, the authors should carry out their experiments in a MHC class I-high IFNGR1ko tumor

model.

Based on their data the authors claim that the anti-tumor effect of Ruxolitinib on IFNGR1ko B16 tumors is dependent on CD4 and CD8 T cells and TNF-alpha. While data shown in Fig. 7A suggest an involvement of CD8 T cells they do not provide evidence for a significant contribution of CD4 T cells. As shown in Fig. 6E, only CD4 T cells are producers of TNF-alpha but not CD8 T cells. This leads to the question about the main cellular source of TNF-alpha. How is the microenvironment of IFNGR1ko B16 tumors composed, which cells are the main producers of TNF-alpha and how does Ruxolitinib affect the activity of those cells?

The authors put a lot of efforts on the molecular characterization of B16 IFNGR1ko cells. They detected elevated levels of pJAK1 and pJAK2 in tumor cells, and linked pJAK1/2 to mTOR signaling. However, they do not provide convincing data that the therapeutic effect of Ruxolitinib on B16 IFNGR1ko tumors is related to the blockade of pJAK1/pJAK2 in tumor cells. Again, does Ruxolitinib affect pJAK levels in other immune cells and affect TNF α production of those cells?

The authors analyzed TCGA melanoma data to demonstrate that IFNGR1-low tumors from melanoma patients showed features similar to B16 IFNGR1ko tumor. They grouped melanomas into IFNGR1-low and IFNGR1-high cohorts and found IFNGR1-low tumors associated with lower levels of T cell infiltration. Here the concern is that IFNGR-high and IFNGR-low are not intrinsic features of the tumor cells. IFNGR1 is an IFN γ -responsive genes and it is more likely, that high levels of IFNGR1 are a consequence of an ongoing anti-tumor T cell response.

Reviewer's Comments:

Reviewer #1 (Remarks to the Author)

In this revised version, the authors clarify several important points regarding the mechanism of constitutive JAK1/2 phosphorylation in IFN γ R1-/- B16 cells. However, they then also show that the effects of Ruxo are not on the cancer cells themselves, but rather on the host immune cells. This represents a reversal of their original conclusion, which is fine – that's what good peer review should occasionally do.

We thank this reviewer for recognizing our efforts to clarify important points regarding the underlying mechanisms. We appreciate the original constructive comments from this reviewer, which were instrumental for the improvement of our study. As this reviewer may recall, our original submission was entitled “JAK inhibition reprograms *intratumoral T cells* and bypasses tumor-intrinsic immunotherapeutic resistance”. So, reprogramming of TILs by Ruxo has always been a main mechanism in our study and this revision is not a reversal of that major line of mechanism. Of note, Ruxo effects depend on the active JAK1/2 in IFN γ R1^{KO} melanomas, as it did not suppress scrambled control melanomas. In other words, activated JAK1/2 in IFN γ R1^{KO} melanoma cells is a prerequisite for Ruxo-mediated reprogramming of TILs.

Perhaps counterintuitively, it seems that blockade of JAK1/2 – critical mediators of IFN γ signaling – in T cells may actually reduce Tregs and increase IFN γ , TNF, and IL-2 production. Apparently (I am extrapolating here somewhat) IFN γ signaling in tumor cells is a critical part of the efficacy of checkpoint blockade, but when tumor-intrinsic IFN γ signaling is rendered obsolete, then the negative effects of IFN γ signaling on T cells outweigh the positive benefits of IFN γ . This is somewhat interesting, but not fully explored in the data shown here.

This is an interesting thought, but it may be limited to tumors lacking IFN- γ signaling, because we reported that T cell IFN- γ signaling is required for ICB therapy of wild-type tumors (2016 *Nat. Commun.*: <https://www.nature.com/articles/ncomms12335>). This can be tested by inoculating mice selectively deficient of IFN γ R1 in T cells with IFN γ R1^{KO} melanoma cells, treated with Ruxo and followed by regular monitoring of tumor growth in the future.

The issue of JAK1/2 signaling in tumor cells remains unresolved. If the increased JAK1/2 is not important for the therapeutic effect of Ruxo, what is it doing? Was this entire exploration of JAK1/2 in tumor cells all for naught? Perhaps the IFN γ R1-/- tumor cells are secreting chemokines or something that is responsible for the baseline alternations in T cell infiltrates. This could be more clearly explained in the Discussion.

As aforementioned, constitutive activation of JAK1/2 in IFN γ R1^{KO} melanoma is a prerequisite for Ruxo efficacy (Fig. 6A&B). We'd like to reiterate that an important goal of our meticulous explorations of activated JAK1/2 in this study (identified through a global kinomics study) was to uncover an effective strategy to bypass ICB resistance of melanomas lacking functional IFN- γ signaling, a major mechanism identified by us and others. Our results convincingly demonstrated that Ruxo is a “targeted” therapy for IFN γ R1^{KO} melanomas. While we largely ruled out a role of altered extrinsic signals (such as secreted cytokines, growth hormones, and extracellular vesicles) in JAK1/2 activation (Fig. 4), we reason that it is the metabolic competition between tumor cells and TILs that modulates TILs in IFN γ R1^{KO} melanoma, which is actively ongoing in my lab. We look forward to sharing that story with the research community, once we have generated compelling data.

“The authors use ref 71 to claim an OS benefit in pancreatic cancer upon treatment with capecitabine and ruxo. This approach unfortunately failed for futility in two Phase III trials (PMID: 29508247). Please delete this sentence and reference.”

We really appreciate this update from this reviewer on results from Phase III trials with capecitabine and Ruxo. We have deleted the sentence and reference.

Reviewer #2 (Remarks to the Author)

The authors have addressed all my concerns; I think that the manuscript deserves publication now. Congratulations for the great work.

We sincerely thank this reviewer for recognizing our diligent efforts to address his/her concerns.

Reviewer #3 (Remarks to the Author)

“Shen et al. aimed to therapeutically target IFN γ signaling-defective melanomas. To do so they generated IFNGR1 ko B16 melanoma cells. B16-IFNGR1ko tumors were non-responsive to anti-CTLA-4 therapy, in contrast to B16 WT tumor. B16-IFNGR1ko tumors showed low-level T cell infiltrates and poor T cell activation. Treatment with Ruxolitinib impaired the growth of IFNGR1ko B16 tumors, and this therapeutic effect was dependent on TNFa.”

We appreciate this concise summary of our key findings by the reviewer.

Comment of the reviewer:

In the first part of their study the authors demonstrated that B16 IFNGR1ko cells do not respond to IFN γ and that IFNGR1ko tumors are no longer sensitive to anti-CTLA-4 therapy. This is expected based on previous studies from the authors and others. In the following they characterized the B16 IFNGR1ko tumor cells, the T cell infiltrate of B16 IFNGR1ko tumors and developed a therapeutic approach against IFN γ -resistant tumors.

It is of strong clinical interest to define strategies to overcome therapy resistance of IFN γ -signaling defective tumors. However, with respect to the study by Shen et al, there are several concerns regarding data interpretation and conclusions.

We thank this reviewer for recognizing the clinical importance of our study, in agreement to the original evaluations of our study.

B16 tumor are characterized by low MHC class I expression, as shown in multiple studies including a recent work by Ribas and colleagues (PMID: 33055240). The efficacy of different T cell-based therapies is dependent on the upregulation of MHC class I expression in B16 tumors which does not take place in IFN γ -resistant tumors. Moreover, it is well established that MHC class I-negative/-low tumors show only low or no T cell infiltrates. Thus, low T cell numbers and low T cell activation in B16 IFNGR1ko tumors could just be a consequence of low MHC class I expression on tumor cells. To address this, the authors should carry out their experiments in a MHC class I-high IFNGR1ko tumor model.

We are aware of the seminal work from Ribas group and many others in this area. In fact, Ribas group and us (together with Pam Sharma and Jim Allison) published the original back-to-back papers in *New England Journal of Medicine* and *Cell* in 2016, which established an essential role of melanoma-intrinsic loss of IFN- γ signaling in ICB resistance. As described in the second paragraph of the Discussion (line 441-458), a wide array of effects can result from tumor cell loss of IFN- γ signaling, including

downregulation of MHC molecules (in a broader term, antigen presentation machinery), downregulation of PD-L1, and regulation of cancer cell stemness/survival/metabolic fitness. It is possible that MHC class I-high IFN γ R1^{KO} tumor model may partially rescue the TIL phenotypes, although we observed a dramatic reduction of MHC class II in IFN γ R1^{KO} melanoma. Future endeavors should be devoted to how those effects act individually or interactively to impart TILs, as all of them can mediate therapeutic resistance.

Based on their data the authors claim that the anti-tumor effect of Ruxolitinib on IFNGR1ko B16 tumors is dependent on CD4 and CD8 T cells and TNF-alpha. While data shown in Fig. 7A suggest an involvement of CD8 T cells they do not provide evidence for a significant contribution of CD4 T cells. As shown in Fig. 6E, only CD4 T cells are producers of TNF-alpha but not CD8 T cells. This leads to the question about the main cellular source of TNF-alpha. How is the microenvironment of IFNGR1ko B16 tumors composed, which cells are the main producers of TNF-alpha and how does Ruxolitinib affect the activity of those cells?

Our previous data showed that blocking CD4 T cells partially abolished therapeutic effects of Ruxo, although statistical significance was not reached due to the limited number of mice used. In response to this reviewer's incisive evaluation, we repeated this experiment with more mice (10) in groups receiving blocking CD4 antibodies and presented the pooled results in the new Fig. 7A. Although not reported in our previous revision, CD8⁺ TILs also produced TNF, but Ruxolitinib did not further increase it (Fig. S7A). Following this reviewer's comment on TNF production by other immune cells, we performed analyses of macrophages (Fig. S7B) and dendritic cells (Fig. S7C) after Ruxo treatment, which did not reveal increased TNF production after Ruxo, either. Although our results revealed a selective increase of TNF production by CD4⁺ TILs after Ruxo, considering the abundant production of TNF by these immune cells (in particular, CD8⁺ TILs and macrophages), we reason that they may also contribute to the overall T cell-dependent anti-tumor responses elicited by Ruxo therapy. Future studies should be performed to track down the cellular source(s) of TNF using a genetic mouse system with conditional TNF deletion in various immune cell populations, which in turn act on TILs, in an autocrine or paracrine manner, to mediate therapeutic effects of Ruxo.

The authors put a lot of efforts on the molecular characterization of B16 IFNGR1ko cells. They detected elevated levels of pJAK1 and pJAK2 in tumor cells, and linked pJAK1/2 to mTOR signaling. However, they do not provide convincing data that the therapeutic effect of Ruxolitinib on B16 IFNGR1ko tumors is related to the blockade of pJAK1/pJAK2 in tumor cells. Again, does Ruxolitinib affect pJAK levels in other immune cells and affect TNF α production of those cells?

In our original submission and revision, we did not examine blockade of p-JAK/STAT in tumor cells after Ruxo treatment, mainly because Ruxo has been widely regarded as a JAK1/2 inhibitor. Following this reviewer's suggestion, we procured reliable flow cytometric antibodies to stain p-JAK2 and p-STAT3 in tumor (CD45⁻) cells after Ruxo therapy. As shown in Fig. S6D, Ruxo did lead to the expected suppression of p-JAK2 and p-STAT3. Unlike tumor cells, TILs are rare in the IFN γ R1^{KO} melanoma. To assess p-JAK2 and p-STAT3 in TILs with great confidence, due to technical limitations, we reason that TILs need to be sorted and pooled from many IFN γ R1^{KO} melanoma-bearing mice. Nevertheless, these results showed that Ruxo was effective at suppressing JAK-STAT pathway in IFN γ R1^{KO} melanoma cells.

The authors analyzed TCGA melanoma data to demonstrate that IFNGR1-low tumors from melanoma patients showed features similar to B16 IFNGR1ko tumor. They grouped melanomas into IFNGR1-low and IFNGR1-high cohorts and found IFNGR1-low tumors associated with lower levels of T cell infiltration. Here the concern is that IFNGR-high and IFNGR-low are not intrinsic features of the tumor cells. IFNGR1 is an IFNg-responsive genes and it is more likely, that high levels of IFNGR1 are a consequence of an ongoing anti-tumor T cell response.

We respectfully disagree with this reviewer on this. IFNGR1 expression on tumor cells has been reported to be mainly regulated by tumor-intrinsic mechanisms such as loss of ELF5–FBXW7 (<https://www.nature.com/articles/s41556-020-0495-y>) and AP-2 α (<https://www.sciencedirect.com/science/article/pii/S0002944011010637>). From our own literature reviewing, we are not aware of upregulation of IFNGR1 by ongoing anti-tumor T cell response or inflammation. Instead, it was reported that type I interferons can downregulate IFN γ R1 in other cellular contexts (myeloid cells) (<https://pubmed.ncbi.nlm.nih.gov/23935197/>). It would be a great help if this reviewer can refer us publications showing that anti-tumor T cell responses directly upregulate IFNGR1 in human tumor cells.

REVIEWERS' COMMENTS

Reviewer #3 (Remarks to the Author):

The authors addressed the major concerns I raised.

Two comments regarding the answers of the authors to my questions:

The authors state: „... As shown in Fig. S6D, Ruxo did lead to the expected suppression of p-JAK2 and p-STAT3“. I guess the authors refer to p-STAT1 since pJAK2 is not shown.

With regard to the TCGA data analyses: although the authors disagree, regulation of IFNGR1 by inflammatory cytokines has been described, as referenced also in the following review PMID: 33862306, by the group of T. Schumacher.

REVIEWERS' COMMENTS

Reviewer #3 (Remarks to the Author):

"The authors addressed the major concerns I raised."

We thank this reviewer for recognizing our efforts. In addition, we are grateful for all the constructive comments from this reviewer that are instrumental to improve our study.

"Two comments regarding the answers of the authors to my questions:"

"The authors state: „... As shown in Fig. S6D, Ruxo did lead to the expected suppression of p-JAK2 and p-STAT3“. I guess the authors refer to p-STAT1 since pJAK2 is not shown."

This may be a misreading or a miscommunication, as Fig. S6D (top panel) in our revision did show suppression of p-JAK2 in tumor cells (CD45⁻) by Ruxo, *in vivo*.

"With regard to the TCGA data analyses: although the authors disagree, regulation of IFNGR1 by inflammatory cytokines has been described, as referenced also in the following review PMID: 33862306, by the group of T. Schumacher."

We thank this reviewer for providing this review article. We agreed to what was stated in this review "IFNGR expression can be modulated by both internal factors and environmental cues, such as TNF and IL1b". Although one of the cited papers in this review (#27) reported that IL-1b can upregulate IFNGR expression in epithelial cells, it remains to be determined which (internal or environmental) cues are more important in regulating IFNGR1 expression in tumor cells.